# DCTdiff: Intriguing Properties of Image Generative Modeling in the DCT Space

**Mang Ning** [1]  **Mingxiao Li** [*2]  **Jianlin Su** [*3]  **Jia Haozhe** [4]  **Lanmiao Liu** [5 1]  **Martin Benes** [6]  **Wenshuo Chen** [4]
**Albert Ali Salah** [1]  **Itir Onal Ertugrul** [1]

## Abstract

This paper explores image modeling from the frequency space and introduces DCTdiff, an end-to-end diffusion generative paradigm that efficiently models images in the discrete cosine transform (DCT) space. We investigate the design space of DCTdiff and reveal the key design factors. Experiments on different frameworks (UViT, DiT), generation tasks, and various diffusion samplers demonstrate that DCTdiff outperforms pixel-based diffusion models regarding generative quality and training efficiency. Remarkably, DCTdiff can seamlessly scale up to 512×512 resolution without using the latent diffusion paradigm and beats latent diffusion (using SD-VAE) with only 1/4 training cost. Finally, we illustrate several intriguing properties of DCT image modeling. For example, we provide a theoretical proof of why 'image diffusion can be seen as spectral autoregression', bridging the gap between diffusion and autoregressive models. The effectiveness of DCTdiff and the introduced properties suggest a promising direction for image modeling in the frequency space. The code is https://github.com/forever208/DCTdiff.

## 1. Introduction

Image discriminative and generative modeling in the RGB space has been the mainstream approach in deep learning for a long time due to the success of Convolutional Neural Networks (Krizhevsky et al., 2012; He et al., 2016) and Vision Transformers (Dosovitskiy et al., 2021). In contrast, images are often stored in a compressed form. For example, JPEG (Wallace, 1991) uses Discrete Cosine Transforma-

---
[*]Equal contribution [1]Utrecht University, the Netherlands [2]KU Leuven, Belgium [3]Moonshot AI, China [4]Shandong University, China [5]Max Planck Institute for Psycholinguistics, the Netherlands [6]University of Innsbruck, Austria. Correspondence to: Mang Ning <m.ning@uu.nl>.

*Proceedings of the 42nd International Conference on Machine Learning*, Vancouver, Canada. PMLR 267, 2025. Copyright 2025 by the author(s).

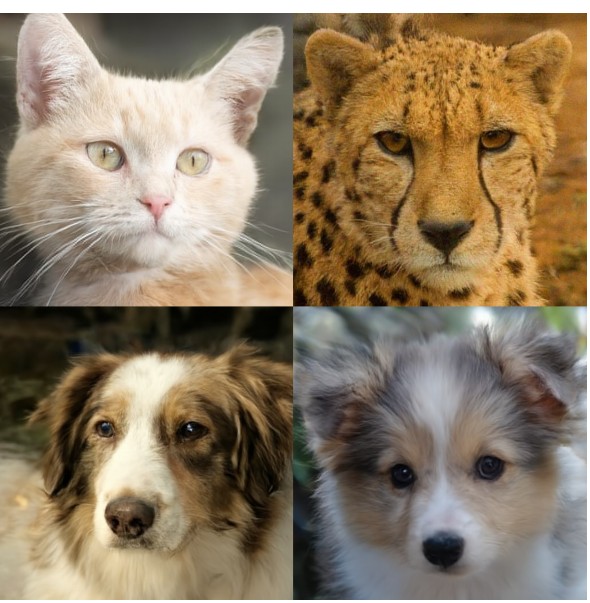

*Figure 1.* Image samples generated by DCTdiff trained on the dataset AFHQ 512×512 without using any VAE.

tion (DCT) and PNG applies DEFLATE (Deutsch, 1996) for compression in a lossy and lossless manner, respectively. In this paper, we explore image modeling in the DCT space with a focus on generative tasks, as they require a complete understanding of the entire image (Goodfellow, 2016). Recently, diffusion models (Song & Ermon, 2019; Ho et al., 2020) have demonstrated remarkable generative performance and been adapted in various tasks, including text-to-image generation (Ramesh et al., 2022; Esser et al., 2024), video generation (Blattmann et al., 2023; Polyak et al., 2024), and 3D synthesis (Poole et al., 2022; Lin et al., 2023). However, diffusion-based generative modeling in the pixel space is expensive and difficult to scale directly to high-resolution generation. Researchers have explored alternatives such as latent space modeling (Rombach et al., 2022) and neural network-based upsampling (Dhariwal & Nichol, 2021) to address these challenges.

We argue that image diffusion modeling in the pixel space is unnecessary due to its inherent redundancy. Instead, we advocate using a (near) lossless compression that provides a compact space for efficient diffusion modeling. JPEG achieves significant image compression by converting pixels

to the DCT frequency domain and eliminating the highest-frequency signals, as they have low energy and are less perceptible to the human eye. Motivated by JPEG, we propose DCTdiff which models the image data distribution entirely in the DCT frequency space. We explore the design space of DCTdiff and uncover the key factors contributing to diffusion modeling in the frequency domain. Through extensive experiments, we demonstrate that DCTdiff surpasses pixel-based diffusion in both generative quality and training efficiency. Crucially, DCT allows for significant lossless compression with negligible computation, enabling DCTdiff to seamlessly scale to $512 \times 512$ image generation without relying on an auxiliary Variational Autoencoder (VAE) (Rombach et al., 2022) which is typically trained with 9 million images and a compound loss.

We further reveal some unique properties of DCT-based image modeling. In Section 5.3, we present a theoretical analysis that frames image diffusion modeling as spectral autoregression. Particularly, the coarse-to-fine autoregressive generation of VAR (Tian et al., 2024) can be summarized as first generating low-frequency signals and then generating high-frequency image details. Also, we highlight that DCT image modeling has the flexibility and advantage of prioritizing different image frequencies according to the granularity of the task. Finally, we introduce a new theorem for image upsampling within the DCT space, offering superior performance over traditional methods such as bilinear or bicubic interpolation. In summary, our contributions are:

- We propose DCTdiff to perform image diffusion modeling in the DCT space for the first time.

- We elucidate the design space of DCTdiff and show that it outperforms the pixel-based and SD-VAE-based latent diffusion models regarding generation quality and training speed.

- We reveal several intriguing properties of image modeling in the DCT space, suggesting its potential for both discriminative and generative tasks and its advantages over conventional pixel-based image modeling.

## 2. Related Work

### 2.1. Diffusion Models

Diffusion models were introduced by Sohl-Dickstein et al. (2015) and improved by Song & Ermon (2019) and Ho et al. (2020). Furthermore, Song et al. (2021b) unify score-based models and denoising diffusion models via stochastic differential equations (SDE), and EDM (Karras et al., 2022) provides a disentangled design space for diffusion models. Recent advancements in diffusion models have been achieved across various dimensions, including classifier guidance (Dhariwal & Nichol, 2021) and classifier-free

guidance (Ho & Salimans, 2022), ODE solver (Lu et al., 2022; Zhou et al., 2024) and SDE solver (Xue et al., 2024), exposure bias (Ning et al., 2023; Li et al., 2024), training dynamics (Karras et al., 2024), model architecture (Peebles & Xie, 2023), noise schedule (Hoogeboom et al., 2023; Hang & Gu, 2024) and sampling schedule (Sabour et al., 2022), sampling variance (Bao et al., 2022), and distillation (Salimans & Ho, 2022; Song et al., 2023). Moreover, Poisson Flow (Xu et al., 2022), Flow Matching (Lipman et al., 2023) and Rectified Flow (Liu et al., 2023) are closely related to the ODE-based diffusion models. Orthogonal to previous studies, we investigate image diffusion modeling from the DCT space for the first time.

### 2.2. Frequency Modeling in Neural Networks

Frequency transformation is often performed as a module of the neural network to speed up the computation (Mathieu et al., 2014; Pratt et al., 2017; Zhang et al., 2018; Tamkin et al., 2020), increase the accuracy of image classification (Fridovich-Keil et al., 2022), or improve the image generative performance (Phung et al., 2023). For example, Huang et al. (2023) applied the Fourier transform to latent representations to adaptively select useful frequencies for target tasks. Yang et al. (2023) incorporate the wavelet-based gating mechanism into the diffusion network to enable dynamic frequency feature extraction. Additionally, DCT was utilized by (Kuzina & Tomczak, 2023) as a low-dimensional latent representation and applied to the VampPrior framework to obtain the flexible prior distribution.

In contrast to treating frequency modeling as a module or auxiliary component of the whole network, researchers have investigated image modeling within the frequency space by transforming image pixels into frequency signals and feeding them to the neural network. For instance, DCTransformer (Nash et al., 2021) proposes generative modeling in the DCT space in an autoregressive manner. Buchholz & Jug (2022) perform image super-resolution tasks in the Fourier domain using an autoregressive model, where low frequencies of an image are conditioned to predict the missing high frequencies. Likewise, Mattar et al. (2024) apply Wavelets Transform to images for autoregressive generation. In addition, Wavelet-Based Image Tokenizer is proposed for image discriminative tasks (Zhu & Soricut, 2024) and generative tasks (Esteves et al., 2024). Recently, JPEG-LM (Han et al., 2024) directly models images and videos as compressed files saved on computers by outputting file bytes in JPEG and AVC formats. In this paper, we adopt DCT space for image generative modeling because DCT concentrates most of the signal's energy into a few low-frequency components (Rao & Yip, 2014), making it very effective for compression. Also, DCT operates on real numbers, simplifying the practical implementation.

## 3. Background

Diffusion models progressively perturb a random data sample $\boldsymbol{x}_0$, drawn from data distribution $P_{\text{data}}$, into a pure noise $\boldsymbol{x}_T$ as time $t$ flows. The forward perturbation process is described by the Stochastic Differential Equation (SDE) (Song et al., 2021b)

$$\mathrm{d}\boldsymbol{x}_t = \boldsymbol{f}(\boldsymbol{x}_t, t)\mathrm{d}t + g(t)\mathrm{d}\boldsymbol{w}_t, \tag{1}$$

where $t \in [0, T]$, $T > 0$ is a constant, $\boldsymbol{f}(\cdot, \cdot)$ and $g(\cdot, \cdot)$ are the drift and diffusion coefficients, and $\boldsymbol{w}_t$ defines the standard Wiener process. A key property of the forward SDE is that there exists an associated reverse-time SDE

$$\mathrm{d}\boldsymbol{x}_t = [\boldsymbol{f}(\boldsymbol{x}_t, t) - g^2(t)\nabla_{\boldsymbol{x}_t}\log p_t(\boldsymbol{x}_t)]\mathrm{d}\bar{t} + g(t)\mathrm{d}\bar{\boldsymbol{w}}_t, \tag{2}$$

where $\mathrm{d}\bar{t}$ represents an infinitesimal negative time step, indicating that this SDE must be solved from $t = T$ to $t = 0$. Moreover, $p_t(\boldsymbol{x}_t)$ denotes the probability distribution of $\boldsymbol{x}_t$, and $\bar{\boldsymbol{w}}_t$ is now a standard Wiener process in the reverse time. This reverse-time SDE results in the same solution $\{\boldsymbol{x}_t\}_{t=0}^{T}$ as the forward SDE (Eq. (1)) (Anderson, 1982), given that $\boldsymbol{x}_T$ is sampled from a prior noise distribution. After training a score model $\boldsymbol{s}_{\boldsymbol{\theta}}(\boldsymbol{x}_t, t) \approx \nabla_{\boldsymbol{x}_t}\log p_t(\boldsymbol{x}_t)$ parameterized by $\boldsymbol{\theta}$ via denoising score matching (Vincent, 2011; Song et al., 2021b), one can plug $\boldsymbol{s}_{\boldsymbol{\theta}}(\boldsymbol{x}_t, t)$ into Eq. (2) to get

$$\mathrm{d}\boldsymbol{x}_t = [\boldsymbol{f}(\boldsymbol{x}_t, t) - g^2(t)\boldsymbol{s}_{\boldsymbol{\theta}}(\boldsymbol{x}_t, t)]\mathrm{d}\bar{t} + g(t)\mathrm{d}\bar{\boldsymbol{w}}_t. \tag{3}$$

Then, we can sample $\boldsymbol{x}_T$ from the prior distribution $\mathcal{N}(\boldsymbol{0}, \boldsymbol{I})$ and solve Eq. (3) backwards in time to obtain the predicted solution trajectory $\{\hat{\boldsymbol{x}}_t\}_{t=0}^{T}$ where $\hat{\boldsymbol{x}}_0$ is viewed as a generated sample from the data distribution $P_{\text{data}}$. Importantly, Song et al. (2021b) reveal that the reverse-time SDE shares the same marginal probability densities $\{p_t(\boldsymbol{x}_t)\}_{t=0}^{T}$ as the *Probability Flow* ODE:

$$\mathrm{d}\boldsymbol{x}_t = [\boldsymbol{f}(\boldsymbol{x}_t, t) - \frac{1}{2}g^2(t)\nabla_{\boldsymbol{x}_t}\log p_t(\boldsymbol{x}_t)]\mathrm{d}t. \tag{4}$$

Again, by replacing the score function $\nabla_{\boldsymbol{x}_t}\log p_t(\boldsymbol{x}_t)$ with the learned score model $\boldsymbol{s}_{\boldsymbol{\theta}}(\boldsymbol{x}_t, t)$, any numerical ODE solver, such as Euler (Song et al., 2021b) and Heun solvers (Karras et al., 2022), can be applied to solve this ODE to obtain an estimated data sample $\hat{\boldsymbol{x}}_0$.

## 4. Design Space of DCTdiff

The DCTdiff proposed in this paper is inspired by the canonical JPEG image codecs (Wallace, 1991). The main idea behind JPEG is that we can achieve data compression by discarding information that is less perceptible to the human eye, especially subtle color variations and high-frequency details, while retaining the essential visual quality of the image. In this paper, we utilize the DCT of JPEG codecs and show that DCT provides a more compact space for image

generative modeling than RGB space in a near-lossless way. The architecture and pipeline of DCTdiff are illustrated in Figure 2 and we now elaborate on each component.

### 4.1. Color Space Transformation and Chroma Subsampling

We follow the JPEG codec and first convert images from the RGB space to the YCbCr color space, containing a brightness component Y (luma) and two color components Cb and Cr (chroma). Formally, given an image $\mathbf{x} \in \mathbb{R}^{h \times w \times 3}$ with height $h$ and width $w$, the color space transformation function can be written as $\mathbf{x}' = \boldsymbol{M}\mathbf{x} + \boldsymbol{b}$, where $\boldsymbol{M} \in \mathbb{R}^{3 \times 3}$ is a fixed transformation matrix and $\boldsymbol{b} \in \mathbb{R}^3$ is the offset vector, and the output $\mathbf{x}' \in \mathbb{R}^{h \times w \times 3}$ represents the YCbCr image. Then we perform 2x chroma subsampling for both Cb and Cr channels since the human eye is more sensitive to brightness than color details (Gonzalez, 2009). As a result, the Y channel of $\mathbf{x}'$ stays the same (denoted as $\mathbf{x}'_{\boldsymbol{y}} \in \mathbb{R}^{h \times w}$), while the Cb and Cr channels become $\mathbf{x}'_{\boldsymbol{cb}} \in \mathbb{R}^{\frac{h}{2} \times \frac{w}{2}}$ and $\mathbf{x}'_{\boldsymbol{cr}} \in \mathbb{R}^{\frac{h}{2} \times \frac{w}{2}}$, respectively (shown in Figure 2 (b)). Note that, chroma subsampling in the YCbCr space brings 2x compression, reducing the signal amount from $3hw$ to $1.5hw$. In Appendix B.1, we empirically show that this 2x compression produced by chroma subsampling significantly accelerates the diffusion training but at the cost of generation quality. In contrast, further transforming the subsampled YCbCr channels to the DCT space improves the quality of generative modeling (Appendix B.2).

### 4.2. 2D Block DCT

After the chroma subsampling, Y, Cb and Cr channels are split into non-overlapping two-dimensional blocks with block size $B$, and the results are denoted as three sets: $\mathbf{x}'_{\boldsymbol{y}} \equiv \{\mathbf{x}_{\boldsymbol{y}}{}^i\}_{i=1}^{4N}, \mathbf{x}'_{\boldsymbol{cb}} \equiv \{\mathbf{x}_{\boldsymbol{cb}}{}^i\}_{i=1}^{N}, \mathbf{x}'_{\boldsymbol{cr}} \equiv \{\mathbf{x}_{\boldsymbol{cr}}{}^i\}_{i=1}^{N}$, where $N$ is the number of blocks in Cb and Cr channels, and $\mathbf{x}_{\boldsymbol{y}}{}^i, \mathbf{x}_{\boldsymbol{cb}}{}^i, \mathbf{x}_{\boldsymbol{cr}}{}^i \in \mathbb{R}^{B \times B}$. Each block is then transformed by a two-dimensional DCT. We use the most common type-II DCT (Ahmed et al., 1974) which converts zero-centered matrix $\boldsymbol{A} \in \mathbb{R}^{B \times B}$ into a DCT block $\boldsymbol{D} \in \mathbb{R}^{B \times B}$ using a series of horizontal and vertical cosine bases:

$$D(u, v) = \alpha(u)\alpha(v) \times$$

$$\sum_{x=0}^{B-1}\sum_{y=0}^{B-1} A(x, y)\cos\left[\frac{(2x+1)u\pi}{2B}\right]\cos\left[\frac{(2y+1)v\pi}{2B}\right] \tag{5}$$

$$\text{where} \quad \alpha(u) = \begin{cases} \sqrt{1/B}, & \text{if} \quad u = 0 \\ \sqrt{2/B}, & \text{if} \quad u \neq 0 \end{cases}$$

The resulting $D(u, v)$ is the DCT coefficient at position $(u, v)$ in the frequency domain, $A(x, y)$ is the YCbCr value at position $(x, y)$ in the spatial domain, $\alpha(u)$ and $\alpha(v)$ are

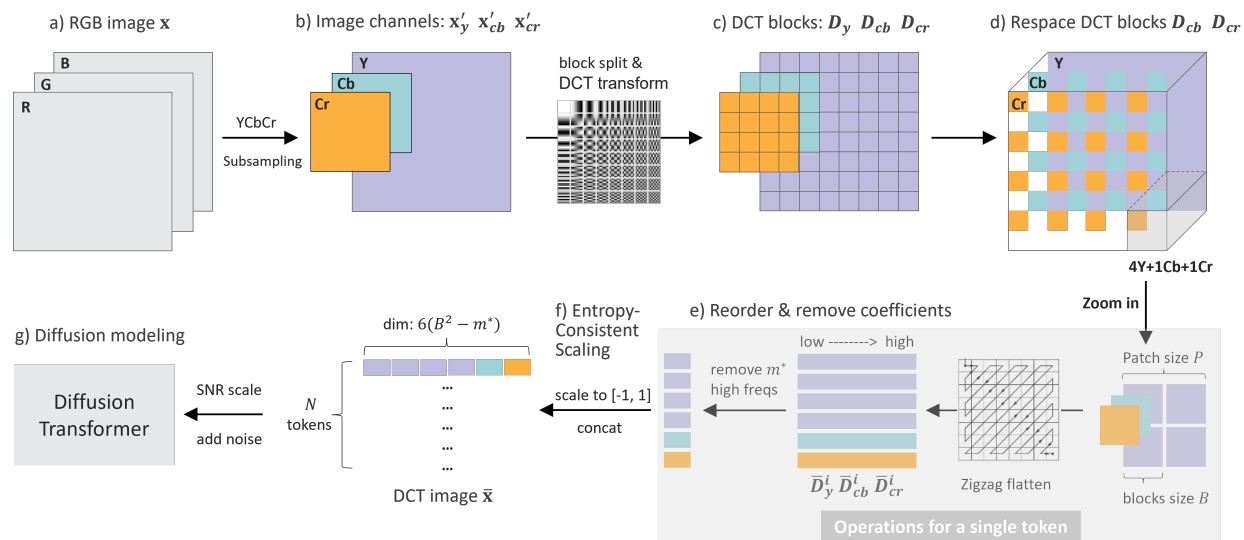

*Figure 2.* The architecture and pipeline of DCTdiff.

normalization factors. We represent the DCT blocks as $\boldsymbol{D_y} \equiv \{\boldsymbol{D_y}^i\}_{i=1}^{4N}$, $\boldsymbol{D_{cb}} \equiv \{\boldsymbol{D_{cb}}^i\}_{i=1}^{N}$, $\boldsymbol{D_{cr}} \equiv \{\boldsymbol{D_{cr}}^i\}_{i=1}^{N}$, corresponding to the DCT outcomes transformed from the Y, Cb and Cr blocks, respectively. JPEG codec uses a fixed block size $B = 8$ due to the trade-off between efficiency and visual quality. However, we find that different image resolutions require different block sizes for optimal diffusion modeling, thus the block size $B$ is set as a variable in DCTdiff. The effect of $B$ is discussed in Appendix B.6.

### 4.3. Frequency Tokenization

We notice that the non-overlapping block operation in DCT resembles the patchfication in Vision Transformer (Dosovitskiy et al., 2021), therefore it is natural to use ViT-based models for DCT diffusion modeling. We propose using UViT (Bao et al., 2023) and DiT (Peebles & Xie, 2023) to construct DCTdiff due to their remarkable performance.

To ensure that the Y, Cb, and Cr blocks within each Transformer token correspond to the same spatial area, we respace the Cb and Cr blocks to expand to the same space as Y blocks (see Figure 2 (d)). Then, four Y blocks, one Cb block, and one Cr block are packed into a single token. Note that, the patch size $P$ in ViT-based models and the block size $B$ in DCTdiff have the relationship $P = 2B$ which will be used throughout the paper. We also tested with other tokenization methods, for instance, each DCT block was considered as a token, but the performance was inferior to the '4Y+1Cb+1Cr' combination.

A notable property of DCT is that many high-frequency coefficients are typically near-zero. These coefficients can be eliminated as they contribute little to the visual quality. To this end, we first turn the two-dimensional DCT

blocks $\boldsymbol{D_y}^i, \boldsymbol{D_{cb}}^i, \boldsymbol{D_{cr}}^i \in \mathbb{R}^{B \times B}$ into one-dimensional vectors $\bar{\boldsymbol{D}}_{\boldsymbol{y}}^i, \bar{\boldsymbol{D}}_{\boldsymbol{cb}}^i, \bar{\boldsymbol{D}}_{\boldsymbol{cr}}^i \in \mathbb{R}^{B^2}$ using the zigzag pattern (shown in Figure 2 (e)), resulting in the coefficients ordered from low-to-high frequency. In order to decide the number of high-frequency coefficients to be chopped off for $\bar{\boldsymbol{D}}_{\boldsymbol{y}}^i, \bar{\boldsymbol{D}}_{\boldsymbol{cb}}^i, \bar{\boldsymbol{D}}_{\boldsymbol{cr}}^i$, we propose the following criteria for generative tasks:

$$m^* = \arg\max_m \{m : \mathrm{rFID}(P_{\mathrm{data}}, P_{\mathrm{dct\_data}}(m)) < \gamma\} \quad (6)$$

where we compute the reconstruction Fréchet Inception Distance (rFID) (Heusel et al., 2017) between the data distribution $P_{\mathrm{data}}$ and the distribution $P_{\mathrm{dct\_data}}$ which is derived from DCT compression by eliminating $m$ high-frequency coefficients. $\gamma$ is a constant and we empirically found that $\gamma = 0.5$ yields a good trade-off between generation quality and compression rate. After removing $m^*$ coefficients, the dimension of $\bar{\boldsymbol{D}}_{\boldsymbol{y}}^i, \bar{\boldsymbol{D}}_{\boldsymbol{cb}}^i, \bar{\boldsymbol{D}}_{\boldsymbol{cr}}^i$ is reduced from $B^2$ to $B^2 - m^*$. Since we concatenate '4Y+1Cb+1Cr' to a token, a Transformer token contains $6(B^2 - m^*)$ frequency coefficients and the number of DCT tokens is $N$. So far, we have transformed the RGB image $\mathbf{x}$ to DCT coefficients, represented as $\bar{\mathbf{x}} \in \mathbb{R}^{N \times 6(B^2 - m^*)}$. Our goal is to model $P_{\mathrm{data}}$ given all DCT samples $\bar{\mathbf{x}}$ using diffusion models.

### 4.4. Diffusion Modeling and Coefficients Scaling

Continuous-time (Song et al., 2021b) and discrete-time diffusion models (Ho et al., 2020) can all be applied for DCTdiff to model $P_{\mathrm{data}}$. For simplicity, we summarize the training process of continuous-time diffusion models and one can refer to Ho et al. (2020) for the details of the discrete-time case. Following Song et al. (2021b), we construct a diffusion process $\{\bar{\mathbf{x}}_t\}_{t=0}^T$ indexed by a continuous time variable

$t \in [0, T]$ such that $\bar{\mathbf{x}}_0 \sim P_{\text{data}}$. We use $P_{0t}(\bar{\mathbf{x}}_t|\bar{\mathbf{x}}_0)$ to denote the perturbation kernel from $\bar{\mathbf{x}}_0$ to $\bar{\mathbf{x}}_t$. Specifically, we follow UViT to employ the Variance Preserving (VP) SDE, so that $P_{0t}(\bar{\mathbf{x}}_t|\bar{\mathbf{x}}_0)$ is a Gaussian:

$$\mathcal{N}(\bar{\mathbf{x}}_t; \bar{\mathbf{x}}_0 e^{-\frac{1}{2} \int_0^t \beta(s)\mathrm{d}s}, \mathbf{I} - \mathbf{I}e^{-\int_0^t \beta(s)\mathrm{d}s}) \quad (7)$$

in which $\beta(.)$ is the noise scale. Then the score model $s_{\boldsymbol{\theta}}(\bar{\mathbf{x}}_t, t)$ is trained by denoising score matching:

$$\mathcal{L}(\boldsymbol{\theta}) = \mathbb{E}_t \lambda(t) \mathbb{E}_{\bar{\mathbf{x}}_0, \bar{\mathbf{x}}_t}[||s_{\boldsymbol{\theta}}(\bar{\mathbf{x}}_t, t) - \nabla_{\bar{\mathbf{x}}_t} \log P_{0t}(\bar{\mathbf{x}}_t|\bar{\mathbf{x}}_0)||_2^2] \quad (8)$$

where $\lambda(t)$ is a positive weighting function, $t$ is sampled from the uniform distribution $U(0, T)$, $\bar{\mathbf{x}}_0 \sim P_{\text{data}}$ and $\bar{\mathbf{x}}_t \sim P_{0t}(\bar{\mathbf{x}}_t|\bar{\mathbf{x}}_0)$. After training the diffusion model, we synthesize images by converting the generated DCT coefficients back to RGB pixels via inverse DCT.

A notable pre-processing in diffusion models is that $\bar{\mathbf{x}}_0$ should be rescaled into the interval $[-1, 1]$ before perturbation. It is trivial for RGB pixels to be shifted and scaled from $[0, 255]$ to $[-1, 1]$. However, we notice that the scaling method affects the diffusion training speed and sample quality when $\bar{\mathbf{x}}_0$ are the DCT coefficients. Different from RGB, the bound of frequency coefficients varies greatly, depending on its position on the spectrum and the channels (Y/Cb/Cr). For instance, the upper bound of the lowest frequency signal in the Y channel (a.k.a DC component) is higher than that of the Cb channel by two orders of magnitude. Therefore, we initially considered a Naive Scaling method: compute the bounds for each frequency and each channel (Y, Cb, Cr) individually, then apply each bound to scale the corresponding frequency coefficients into $[-1, 1]$, respectively. However, we observe that Naive Scaling broadens the distribution of high-frequency coefficients, leading to slow training and low generated sample quality (see Appendix A.1). To maintain the shape of the original probability density, we thus propose the Entropy-Consistent Scaling approach in which all frequency signals are scaled by the bound of the DC component ($D(0, 0)$ of Y blocks) since it yields the largest bound. To avoid the influence of extreme values, we compute the bound $\eta \in \mathbb{R}$ within $\tau$ percentile and $100 - \tau$ percentile:

$$\eta = \max(|P_\tau|, |P_{100-\tau}|) \quad (9)$$

where $P_\tau$ denotes the $\tau$th percentile of the DC component distribution. We empirically find that $\tau = 98.25$ yields the best performance of DCTdiff among all datasets. Hereafter, we assume that $\bar{\mathbf{x}}_0$ has been scaled by $\eta$, namely $\bar{\mathbf{x}}_0 = \bar{\mathbf{x}}_0/\eta$.

**4.5. SNR Scaling**

An inherent property of DCT is that most of the signal's energy is compacted into a few low-frequency components, so that high-frequency components are near zero and quickly

destroyed by the noise term during the forward diffusion process (detailed discussion in Section 5.3). Consequently, this results in the phenomenon that perturbing an image in the DCT space is faster than perturbing an image in pixel space, despite using the same forward SDE (see Figure 7). Furthermore, the larger the block size $B$, the more the energy is concentrated in low frequencies, thus the faster the forward perturbation is. We can also infer this phenomenon from the bound $\eta$ mentioned above. To this end, we introduce a brief corollary concerning $\eta$: given a dataset, $\eta$ doubles if the block size $B$ is doubled.

*Sketch of Proof.* Since $\eta$ is derived from $D(0, 0)$, we investigate how $D(0, 0)$ changes under block size $2B$ and $B$. Plugging $2B$ and $B$ into Eq. (5) yields

$$\frac{1}{2B} \sum_{x=0}^{2B-1} \sum_{y=0}^{2B-1} A(x, y) = 2 \times \frac{1}{B} \sum_{x=0}^{B-1} \sum_{y=0}^{B-1} A(x, y)$$

if we assume that $A(x, y)$ has the same mean within the $2B \times 2B$ block and $B \times B$ block. Thus $D(0, 0)$ doubles when $B$ is doubled, causing $\eta$ to increase twofold. Recall that we scale down all frequency coefficients by $\eta$ before adding noise, a larger $\eta$ would result in more coefficients close to zero and destroyed by the noise in the early stage of forward diffusion. To counteract the effect of block size, we propose to scale the SNR (Signal-Noise-Ratio) of the default noise schedule (inspired by Hoogeboom et al. (2023)). We leave the derivation and implementation of SNR Scaling to Appendix A.2. Experiments show that SNR Scaling improves the sample quality without affecting the training convergence (Appendix B.5).

## 5. Intriguing Properties of Image Modeling in the DCT Space

### 5.1. Frequency Prioritization

Recall that training the score model is equivalent to predicting the isotropic Gaussian noise added on the clear image given the noisy image (Dhariwal & Nichol, 2021). Intuitively, the task is to reconstruct each frequency coefficient in $\bar{\mathbf{x}}_0$ or reconstruct each pixel in the case of RGB image $\mathbf{x}_0$. Since we cannot say which pixel is more important than another pixel, the training objective (Eq. (8)) treats every pixel equally. However, an intriguing property of DCT coefficients is that a low-frequency signal typically contributes more to the image quality than a high-frequency signal. Meanwhile, we observe that the lower the frequency of the signal, the larger the entropy of its distribution. Thereby, we can prioritize the modeling of low-frequency signals of $\bar{\mathbf{x}}_0$ by adding Entropy-Based Frequency Reweighting (EBFR) into eq. (8), leading to $\mathcal{L}_{EBFR}(\boldsymbol{\theta})$:

$$\mathbb{E}_t \lambda(t) \mathbb{E}_{\bar{\mathbf{x}}_0, \bar{\mathbf{x}}_t}[\boldsymbol{H}(B)||s_{\boldsymbol{\theta}}(\bar{\mathbf{x}}_t, t) - \nabla_{\bar{\mathbf{x}}_t} \log P_{0t}(\bar{\mathbf{x}}_t|\bar{\mathbf{x}}_0)||_2^2] \quad (10)$$

where $\boldsymbol{H}(B) \in \mathbb{R}^{3(B^2-m^*)}$ is the entropy vector of the valid $3(B^2 - m^*)$ frequency distributions. Given a dataset, $\boldsymbol{H}(B)$ only depends on the block size $B$ once $m^*$ is fixed. We empirically show that EBFR improves the sample quality without affecting the training speed (see Appendix B.4).

The frequency prioritization strategy can be easily extended to discriminative tasks using prior knowledge. Specifically, we can allocate more network capacity to model high-frequency inputs on tasks requiring a good understanding of fine details, for example, text and handwriting recognition, medical image analysis (Ronneberger et al., 2015), fingerprint recognition, forgery detection (Wu et al., 2019), etc. In contrast, we can explicitly highlight the low-frequency signals on tasks focusing on general shapes and overall structures, for instance, scene recognition (Zhou et al., 2017), object detection in natural scenes (Redmon, 2016), action recognition (Simonyan & Zisserman, 2014), and so on.

### 5.2. Significant Lossless Compression under DCT

Unlike the Fourier transform, DCT operates on real numbers using cosine functions, which effectively match the even symmetric extension of a signal (Rao & Yip, 2014). This alignment with signal characteristics allows the DCT to represent an image or other signals using fewer frequency coefficients (mostly the low-frequency ones). For generative tasks, we can again use rFID($P_{\text{data}}, P_{\text{dct\_data}}(m)$) to measure the information loss when removing $m$ high-frequency coefficients. Table 1 presents the results of rFID using 50k images from the data distribution $P_{\text{data}}$. If we consider $\gamma = 0.5$ as a lossless compression for image generation, DCT could achieve $4\times$ compression on $256{\times}256$ images and $7.11\times$ compression on $512{\times}512$ images. In Section 6.1, we will show that the significant compression of DCT enables the diffusion model to scale smoothly up to high-resolution generation, while pixel diffusion fails due to its high dimensionality. We also visually compare the image quality of VAE compression (Rombach et al., 2022) and DCT compression. As a training-free, computationally negligible, and domain insensitive compression method, DCT retains more image details than VAE compression (Figure 9).

*Table 1.* rFID($P_{\text{data}}, P_{\text{dct\_data}}(m)$) when removing $m$ coefficients on the dataset FFHQ $256{\times}256$ and FFHQ $512{\times}512$. The compression ratio is relative to the RGB image having $3*wh$ signals.

| Dataset | Block size | $m$ | rFID | Compression ratio |
|---------|-----------|-----|------|-------------------|
| FFHQ 256×256 | 4 | 7 | 0.19 | 3.56 |
| | | **8** | **0.49** | **4.00** |
| | | 9 | 0.96 | 4.57 |
| FFHQ 512×512 | 8 | 44 | 0.23 | 6.40 |
| | | **46** | **0.48** | **7.11** |
| | | 48 | 1.18 | 8.00 |

### 5.3. Image Diffusion Is Spectral Autoregression

Recently, Dieleman (2024) has empirically shown that pixel-based diffusion models perform approximate autoregression in the frequency domain. Intuitively, diffusion models destroy an image's high-frequency signals and then progressively destroy lower-frequency signals as time $t$ flows in the forward diffusion process (Yang et al., 2023; Kingma & Gao, 2023; Rissanen et al., 2023). In this paper, we provide the theoretical proof for this phenomenon.

**Theorem 5.1.** *Consider a diffusion model described by* $\mathrm{d}\boldsymbol{x}_t = \boldsymbol{f}(\boldsymbol{x}_t, t)\mathrm{d}t + g(t)\mathrm{d}\boldsymbol{w}_t$. *Let $\omega$ denote the frequency,* $\hat{\boldsymbol{x}}_0(\omega)$ *and* $\hat{\boldsymbol{x}}_t(\omega)$ *represent the Fourier transform of the pixel image $\boldsymbol{x}_0$ and $\boldsymbol{x}_t$, respectively. The averaged power spectral density of the noisy image $\boldsymbol{x}_t$ satisfies:*

$$\mathbb{E}\left[|\hat{\boldsymbol{x}}_t(\omega)|^2\right] = |\hat{\boldsymbol{x}}_0(\omega)|^2 + \int_0^t |g(s)|^2 \mathrm{d}s \qquad (11)$$

*in which $|\hat{\boldsymbol{x}}_0(\omega)|^2$ is the power spectral density of the image $\boldsymbol{x}_0$ and natural images have the power-law:* $|\hat{\boldsymbol{x}}_0(\omega)|^2 = K|\omega|^{-\alpha}$ *(Ruderman, 1997)($K$ and $\alpha$ are constants). Meanwhile, $\int_0^t |g(s)|^2 \mathrm{d}s$ is independent of frequency $\omega$ and appears as a horizontal line in the spectral density graph.*

*Sketch of Proof.* Taking the integral of the forward diffusion SDE yields $\boldsymbol{x}_t = \boldsymbol{x}_0 + \int_0^t g(s)\mathrm{d}\boldsymbol{w}_s$ (assuming $\boldsymbol{f}(\boldsymbol{x}_t, t) = 0$ for VE-SDE). Since Fourier transform is linear, we have $\hat{\boldsymbol{x}}_t(\omega) = \hat{\boldsymbol{x}}_0(\omega) + \hat{\boldsymbol{\epsilon}}_t(\omega)$ in the frequency domain, where $\hat{\boldsymbol{\epsilon}}_t(\omega)$ is the Fourier transform of the noise term $\int_0^t g(s)\mathrm{d}\boldsymbol{w}_s$. By taking the expectation over the Wiener process $\boldsymbol{w}_s$, we can obtain $\mathbb{E}\left[|\hat{\boldsymbol{x}}_t(\omega)|^2\right] = |\hat{\boldsymbol{x}}_0(\omega)|^2 + \mathbb{E}\left[|\hat{\boldsymbol{\epsilon}}_t(\omega)|^2\right]$ due to $\mathbb{E}\left[|\hat{\boldsymbol{\epsilon}}_t(\omega)|\right] = 0$. According to Itô isometry (Itô, 1944), we have $\mathbb{E}[|\hat{\boldsymbol{\epsilon}}_t(\omega)|^2] = \int_0^t |g(s)|^2 ds$ which leads to Eq. (11) (Please refer to Appendix A.3 for the detailed proof).

In Eq. (11), $|\hat{\boldsymbol{x}}_0(\omega)|^2$ quickly decreases to near-zero as frequency $\omega$ increases. So, the spectral density of the high-frequency component, $\mathbb{E}\left[|\hat{\boldsymbol{x}}_t(\omega)|^2\right]$, is mainly decided by the noise term $\int_0^t |g(s)|^2 \mathrm{d}s$. Since the noise term is monotonically increasing as $t$ grows from $0 \to T$, we can see that the noise term in the forward diffusion SDE first mainly destroys the high-frequency component of image $\boldsymbol{x}_0$, and then gradually diminishes the lower-frequency signals. For every frequency $\omega$, we can further determine the required time to reach a specific SNR, see Appendix A.3 for the derivation. Interestingly, VAR proposed to generate images from coarse to fine by predicting the next-resolution image (Tian et al., 2024). We believe the success of VAR stems from the 'spectral autoregression' property of images. More recently, Yu et al. (2025) and Huang et al. (2025) have shown the efficacy of explicit frequency autoregression for image generative modeling.

Note that Theorem 5.1 holds if we replace the Fourier trans-

form with DCT since DCT is also a linear transformation and is a simplified, real-valued variant of the Fourier transform. Inspired by (Dieleman, 2024), we visualize the averaged power spectral density from the DCT space (detailed in Appendix A.4). The resulting curves in Figure 6 resemble the case of the Fourier Transform, indicating that pixel diffusion is also spectral autoregression in the DCT space.

Similar to pixel diffusion, a frequency-based diffusion process (e.g. DCTdiff) simultaneously adds isotropic noise to the whole spectrum in the forward diffusion process, i.e., equally perturbing high-frequency and low-frequency signals at each time step. As high frequencies have low energy, they are first corrupted by the noise. Thereby, we conclude that frequency-based image diffusion is also spectral autoregression. However, DCT concentrates the image energy into low frequencies (Rao & Yip, 2014), leaving most high-frequency components close to zero, so that DCT exhibits a fast 'noise-adding' forward diffusion process (see Figure 7), which motivated the proposal of SNR Scaling method (discussed in Section 4.5).

### 5.4. DCT Upsampling Outperforms Pixel Upsampling

In the aspect of image processing, we find that upsampling in the DCT space produces higher-quality images than upsampling in the pixel space (e.g., using bilinear or bicubic interpolation). Motivated by Dugad & Ahuja (2001), we introduce the following theorem to relate the frequency between low-resolution and high-resolution images in the DCT space given any DCT block size $B$.

**Theorem 5.2.** *Let $A \in \mathbb{R}^{2B \times 2B}$ be a matrix representing an image, and define $\bar{A} \in \mathbb{R}^{B \times B}$ as the matrix obtained by average pooling of $A$, where each element is computed as:*

$$\bar{A}(i,j) = \frac{1}{4} \sum_{m=0}^{1} \sum_{n=0}^{1} A(2i + m, 2j + n).$$

*Suppose $D \in \mathbb{R}^{2B \times 2B}$ represents the DCT of $A$ under block size $2B$ and $\bar{D} \in \mathbb{R}^{B \times B}$ represents the DCT of $\bar{A}$ under block size $B$. Then, for $k, l \in \{0, 1, \dots, B - 1\}$, the elements of $\bar{D}$ can be approximated by:*

$$\bar{D}(k,l) \approx \frac{1}{2} \cos\left(\frac{k\pi}{4B}\right) \cos\left(\frac{l\pi}{4B}\right) D(k,l), \quad (12)$$

*where $(k, l)$ indexes the elements of the matrices $D$ and $\bar{D}$. Appendix A.5 provides the full proof.*

Based on Theorem 5.2, we propose the DCT Upsampling algorithm. For each DCT block $\bar{D}$ converted from a low-resolution image, the algorithm computes $D(k,l)$ from $\bar{D}(k,l)$ according to Eq. (12), generating the low-frequency coefficients (purple block in Figure 3) of $D$. For the remaining frequency coefficients in $D$, we fill them up with zeros

since they are near-zero in practice. The resulting $D$ can be converted back to the pixel space to create a high-resolution image.

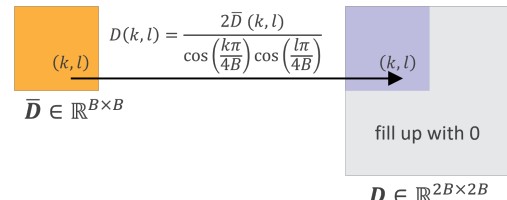

*Figure 3.* Illustration of the DCT Upsampling algorithm.

We evaluate DCT Upsampling both qualitatively and quantitatively. We show an example in Figure 10 to illustrate the difference between Pixel Upsampling by bicubic interpolation (Gonzalez, 2009) and DCT Upsampling. The latter alleviates the blurry effect and exhibits an improved image quality. Also, we apply FID to evaluate the distance between the ground truth data distribution and the upsampled data distribution. Experimentally, DCT Upsampling achieves FID 9.79, outperforming Pixel Upsampling (FID 12.53). One application of DCT Upsampling is super-resolution image generation. Specifically, one can follow the paradigm of ADM (Dhariwal & Nichol, 2021): first train a low-resolution image generation model, then replace the bilinear interpolation with our DCT Upsampling to obtain a better draft high-resolution image, and another diffusion model can finally refine this draft image. Our experimental results and discussion are presented in Appendix A.10.

## 6. Experiments

To evaluate the performance of DCTdiff, we construct the models based on UViT and DiT without changing their Transformer architectures and compare DCTdiff with these two base models. The metrics used for the comprehensive comparison are FID (Heusel et al., 2017), training cost, inference speed, and scalability. Other evaluation metrics (e.g., recall and precision) are presented in Appendix A.7. To ensure fairness, we always use the same model size, patch size, and training parameters for DCTdiff and the base model (unless otherwise noted). The complete network parameters and training settings are listed in Appendix A.6, in which we also elaborate on the choice of DCT parameters and show that determining these parameters is effortless for a new dataset. We leave all ablation studies to Appendix B.

### 6.1. Results on UViT

UViT (Bao et al., 2023) utilizes the continuous-time diffusion framework and different solvers for sampling. We train both UViT and DCTdiff from scratch using the default training parameters suggested by UViT. The datasets include CIFAR-10 (Krizhevsky et al., 2009), CelebA 64 (Liu et al.,

*Table 2.* FID-50k of UViT and DCTdiff using DDIM sampler and DPM-Solver under different NFEs. We implement class-conditional generation on ImageNet 64, and unconditional generation on the rest of the datasets.

| NFE | Model | Euler ODE solver (DDIM sampler) | | | | DPM-Solver | | | |
|---|---|---|---|---|---|---|---|---|---|
| | | CIFAR-10 | CelebA 64 | ImageNet 64 | FFHQ 128 | CIFAR-10 | CelebA 64 | ImageNet 64 | FFHQ 128 |
| 100 | UViT | 6.23 | 1.99 | 10.65 | 13.87 | 5.80 | **1.57** | 10.07 | 9.18 |
| | DCTdiff | **5.02** | **1.91** | **8.69** | **8.22** | **5.28** | 1.71 | **9.73** | **6.25** |
| 50 | UViT | 7.88 | 3.50 | 15.05 | 26.26 | 5.82 | **1.58** | 10.09 | 9.20 |
| | DCTdiff | **5.21** | **2.24** | **8.70** | **9.99** | **5.30** | 1.72 | **9.78** | **6.28** |
| 20 | UViT | 21.48 | 31.09 | 52.10 | 87.68 | 6.19 | **1.73** | 10.25 | 9.21 |
| | DCTdiff | **6.81** | **3.84** | **21.88** | **24.88** | **5.54** | 1.84 | **9.85** | **7.29** |
| 10 | UViT | 81.67 | 224.21 | 166.63 | 209.69 | 26.65 | **4.37** | 13.27 | 14.26 |
| | DCTdiff | **12.45** | **67.78** | **129.93** | **161.05** | **9.10** | 5.29 | **12.38** | **12.87** |

2015), ImageNet 64 (Chrabaszcz et al., 2017), FFHQ 128, FFHQ 256, FFHQ 512 (Karras et al., 2019) and AFHQ 512 (Choi et al., 2020). We perform class-conditional generation on ImageNet 64 and unconditional generation for the other datasets. We test the sample quality using FID-50k under different Number of Function Evaluation (NFE) and two ODE solvers: DDIM sampler (Song et al., 2021a) and DPM-Solver (Lu et al., 2022).

Results on Table 2 show that DCTdiff consistently and sometimes dramatically outperforms UViT regardless of NFEs and solvers (except for the outlier of CelebA 64 using DPM-Solver), demonstrating the effectiveness of image diffusion modeling in the DCT space. We believe the outlier can be attributed to the UViT training parameters being highly suited to CelebA 64, since the results of CelebA 64 based on DiT still demonstrate the superiority of DCTdiff.

Note that UViT uses SD-VAE to perform latent diffusion (Rombach et al., 2022) when the image resolution reaches 256×256 because pixel diffusion modeling in the high-dimensional space is difficult (we attempted to train the UViT model directly in the pixel space on FFHQ 256, resulting in FID=120). However, we show that the diffusion paradigm of DCTdiff can be easily scaled to 512×512 image generation without VAE. We denote the UViT using SD-VAE as UViT (latent) and compare it with our DCTdiff on three datasets. Table 3 indicates that DCTdiff achieves competitive FID to UViT (latent) on FFHQ 256 and outperforms UViT (latent) when the resolution rises to 512.

### 6.2. Results on DiT

DiT (Peebles & Xie, 2023) applies the discrete-time diffusion framework and the Euler SDE solver (DDPM sampler) for image sampling. To verify the generalization of DCTdiff, we utilize DiT as the baseline, and train DiT (in the pixel space) and DCTdiff from scratch with the same training settings on CelebA 64 and FFHQ 128 datasets. The resulting FIDs in Table 4 show that DCTdiff surpasses DiT under different sampling steps regarding the generation quality.

*Table 3.* FID-50k of UViT (latent) and DCTdiff on high-resolution image datasets for unconditional generation. DPM-Solver is used for sampling.

| NFE | Model | Dataset | | |
|---|---|---|---|---|
| | | FFHQ 256 | FFHQ 512 | AFHQ 512 |
| 100 | UViT (latent) | **4.26** | 10.89 | 10.86 |
| | DCTdiff | 5.08 | **7.07** | **8.76** |
| 50 | UViT (latent) | **4.29** | 10.94 | 10.86 |
| | DCTdiff | 5.18 | **7.09** | **8.87** |
| 20 | UViT (latent) | **4.74** | 11.31 | 11.94 |
| | DCTdiff | 6.35 | **8.04** | **10.05** |
| 10 | UViT (latent) | 13.29 | 23.61 | 28.31 |
| | DCTdiff | **12.05** | **19.67** | **21.05** |

Importantly, we find that the parameters $(B, \tau, m^*, c)$ of DCTdiff only depend on the image resolution, and are invariant to the base models and dataset types (Appendix A.6), making the application of DCTdiff convenient in practice.

*Table 4.* FID-50k of DiT and DCTdiff using DDPM sampler under different NFEs for unconditional generation.

| NFE | Model | Dataset | |
|---|---|---|---|
| | | CelebA 64 | FFHQ 128 |
| 100 | DiT | 5.11 | 12.81 |
| | DCTdiff | **3.84** | **11.16** |
| 50 | DiT | 8.17 | 18.44 |
| | DCTdiff | **6.23** | **15.23** |
| 20 | DiT | 15.64 | 33.56 |
| | DCTdiff | **12.96** | **25.59** |
| 10 | DiT | 24.76 | 49.64 |
| | DCTdiff | **20.87** | **43.14** |

### 6.3. Training Cost and Inference Speed

In addition to sample quality, we also compare the training costs between pixel-based UViT and DCTdiff. Results in Table 5 demonstrate that the training of DCTdiff is faster

than that of UViT with the training acceleration up to 2.5×. When comparing UViT (latent) with DCTdiff, we follow Peebles & Xie (2023) and leverage GFLOPs (Giga Floating Point Operations) to indicate the computational complexity required for a single network-forward-pass. We find that the GFLOPs of SD-VAE are huge and increase dramatically when the resolution rises from 256×256 to 512×512. As a result, the total training cost of DCTdiff on FFHQ 256 is comparable to that of UViT (latent), but the training cost of DCTdiff on AFHQ 512 is only one-quarter of that of UViT (latent). We attribute the efficient training of DCTdiff to the compact space provided by DCT.

Table 5. Training cost of UViT and DCTdiff. We use the same batch size for the two models on each dataset. The training cost is indicated by GFLOPs and Training steps (convergence).

| Dataset | Model | # Parameters | GFLOPs | Training steps |
|---|---|---|---|---|
| CelebA 64 | UViT | 44M | 11 | 400k |
| | DCTdiff | 44M | 11 | **250k** |
| FFHQ 128 | UViT | 44M | 11 | 750k |
| | DCTdiff | 44M | 11 | **300k** |
| FFHQ 256 | UViT (latent) | 131M + 84M | 169 | **200k** |
| | DCTdiff | 131M | **133** | 300k |
| AFHQ 512 | UViT (latent) | 131M + 84M | 575 | 225k |
| | DCTdiff | 131M | **133** | **225k** |

We further evaluate the inference speed by measuring the wall-clock time for both DCTdiff and UViT. Since DCTdiff and the pixel-based UViT share the same network parameters and GFLOPs, they exhibit the same inference times. However, differences arise when comparing DCTdiff and latent UViT at resolutions of 256×256 and 512×512: the SD-VAE component in latent UViT incurs high computational cost, whereas DCTdiff consumes more Transformer tokens. As shown in Table 6, DCTdiff is faster than latent UViT in low-NFE settings but is slower at high-NFE conditions (without considering the sampling quality). Notably, DCTdiff demonstrates clear advantages over UViT in terms of inference time when achieving comparable generation quality (refer to Appendix A.9).

Table 6. Wall-clock inference time on AFHQ 512×512. We generate 10k samples using one A100 GPU. Inference GFLOPs is appended in the brackets.

| Model | NFE | | | |
|---|---|---|---|---|
| | 100 | 50 | 20 | 10 |
| UViT (latent) | 20.2 min (4640) | 13.4 min (2940) | 9.3 min (1920) | 7.9 min (1580) |
| DCTdiff | 47.8 min (13300) | 23.9 min (6650) | 9.6 min (2660) | 4.8 min (1330) |

## 6.4. Scalability of DCTdiff

A key advantage of diffusion Transformers is their network scalability (Peebles & Xie, 2023). We empirically demonstrate that DCTdiff inherits this property and achieves improved sample quality as network capacity increases. Tables 7 and 8 report the scalability results on the datasets CIFAR-10 and FFHQ 128, respectively, where DCTdiff consistently outperforms UViT in terms of FID. A similar scaling effect of DCTdiff is observed on the dataset ImageNet 64 as well (Appendix A.8).

Table 7. Scalability of UViT and DCTdiff on CIFAR-10 (patch size=4). FID-50k is reported using DDIM sampler.

| Model | Dpeth | # Para | NFE | | |
|---|---|---|---|---|---|
| | | | 100 | 50 | 20 |
| UViT (small) | 12 | 44M | 7.36 | 8.45 | 21.18 |
| DCTdiff (small) | | | **6.51** | **6.62** | **7.87** |
| UViT (mid) | 16 | 131M | 6.23 | 7.88 | 21.48 |
| DCTdiff (mid) | | | **5.02** | **5.21** | **6.81** |
| UViT (mid, deep) | 20 | 161M | 6.05 | 7.33 | 20.27 |
| DCTdiff (mid, deep) | | | **4.25** | **4.54** | **5.96** |

Table 8. Scalability of UViT and DCTdiff on FFHQ 128. FID-50k is reported using DPM-Solver.

| Model | Dpeth | # Para | NFE | | |
|---|---|---|---|---|---|
| | | | 100 | 50 | 20 |
| UViT (small) | 12 | 44M | 9.18 | 9.20 | 9.21 |
| DCTdiff (small) | | | **6.25** | **6.28** | **7.29** |
| UViT (mid) | 16 | 131M | 6.96 | 6.99 | 7.40 |
| DCTdiff (mid) | | | **5.13** | **5.20** | **6.19** |
| UViT (mid, deep) | 18 | 146M | 6.05 | 6.06 | 6.27 |
| DCTdiff (mid, deep) | | | **4.98** | **5.05** | **5.94** |

## 7. Conclusion

In this paper, we explore image generative modeling in the DCT space and propose DCTdiff which shows superior performance over pixel-based and latent diffusion models. In particular, we reveal several interesting properties of image modeling from the DCT space, suggesting a promising research direction for image discriminative and generative tasks. However, the frequency-oriented Transformer architecture and image generation in 1024×1024 resolution are not explored in this paper, which encourages future study. Additionally, considering the inherent high temporal redundancy, video compression and modeling in the frequency space also hold potential for future exploration.

## Impact Statement

This paper advances the field of image generative modeling through the DCT frequency space. Similar to other diffusion-based and autoregressive models, our work has applications in text-to-image generation and image editing over various domains (art, education, social media, and so on). There are many potential societal consequences of our work, none of which we feel must be specifically highlighted here.

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

# A. Appendix

## A.1. Naive Scaling and Entropy-Consistent Scaling

We show the bounds of Naive Scaling and Entropy-Consistent Scaling in Algorithm 1 and Algorithm 2. Given a dataset, we randomly draw 50,000 images and convert each image into DCT blocks $\{D_y{}^i\}_{i=1}^{4N}, \{D_{cb}{}^i\}_{i=1}^{N}, \{D_{cr}{}^i\}_{i=1}^{N}$. The total Y blocks $D_y \in \mathbb{R}^{200000N \times B^2}$, Cb blocks $D_{cb} \in \mathbb{R}^{50000N \times B^2}$, and Cr blocks $D_{cr} \in \mathbb{R}^{50000N \times B^2}$ are used for Monte Carlo estimation of the bounds ($\bar{\eta}$) of Naive Scaling. We use only Y blocks $D_y \in \mathbb{R}^{200000N \times B^2}$ to estimate the bound ($\eta$) of Entropy-Consistent Scaling given block size $B$ and percentile $\tau$.

---

**Algorithm 1** Bound of Naive Scaling

1: **Given** $B, \tau, D_y, D_{cb}, D_{cr}$
2: Initialize $\bar{\eta} = list()$
3: **for** $x := D_y, D_{cb}, D_{cr}$ **do**
4:    **for** $i := 0, 1, ...B^2 - 1$ **do**
5:       $up = np.percentile(x[:, i], \tau)$
6:       $low = np.percentile(x[:, i], 100 - \tau)$
7:       **if** $|low| > |up|$ **then**
8:          $\bar{\eta}.append(|low|)$
9:       **else**
10:         $\bar{\eta}.append(|up|)$
11:       **end if**
12:    **end for**
13: **end for**
14: **return** $\bar{\eta}$

---

**Algorithm 2** Bound of Entropy-Consistent Scaling

1: **Given** $B, \tau, D_y$
2: Initialize $\eta$
3: $x \leftarrow D_y[:, 0]$
4: $up = np.percentile(x, \tau)$
5: $low = np.percentile(x, 100 - \tau)$
6: **if** $|low| > |up|$ **then**
7:    $\eta \leftarrow |low|$
8: **else**
9:    $\eta \leftarrow |up|$
10: **end if**
11: **return** $\eta$

---

In Figure 4, we illustrate the difference between applying Naive Scaling and Entropy-Consistent Scaling using $B = 2$ and $\tau = 97$ on CelebA 64 dataset. The first row of Figure 4 displays the distributions of DCT coefficients $(D(0, 0), D(0, 1), D(1, 0), D(1, 1))$ before scaling. It is clear that Naive Scaling increases the entropy of the original distributions of $D(0, 1), D(1, 0)$ and $D(1, 1)$ while Entropy-Consistent Scaling preserves the entropy.

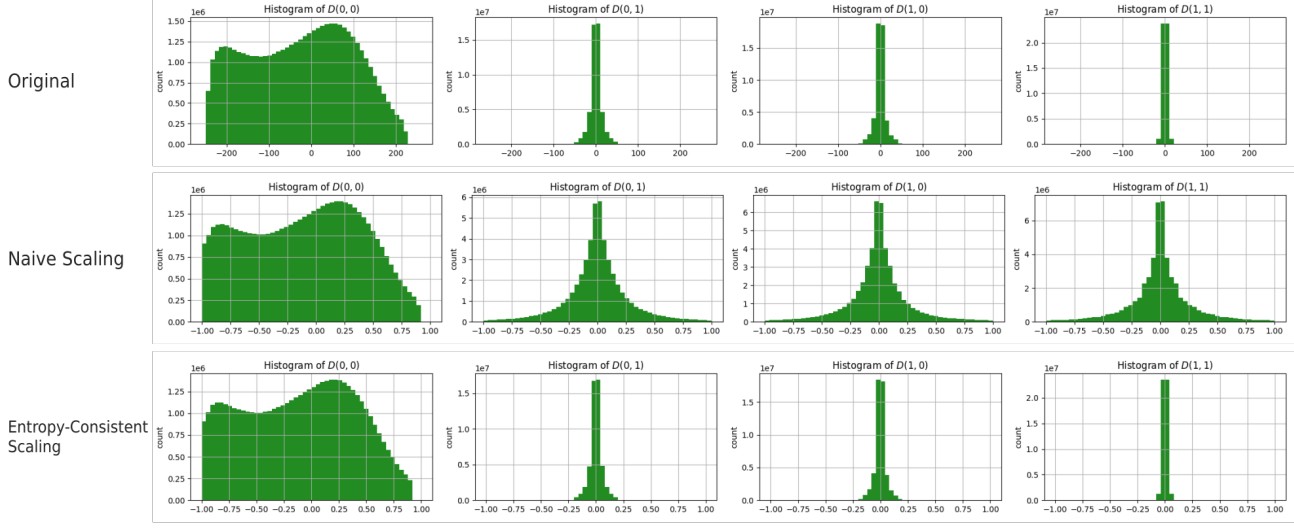

*Figure 4.* Histograms of DCT coefficients before scaling and after Naive Scaling and Entropy-Consistent Scaling.

## A.2. SNR Scaling in Continuous-time and Discrete-time Diffusion Models

### A.2.1. SNR SCALING IN CONTINUOUS-TIME DIFFUSION MODELS

Following UViT (Bao et al., 2023), we use VP-SDE for the continuous-time diffusion model which has the forward perturbation kernel:

$$\mathcal{N}(\bar{\mathbf{x}}_t; \bar{\mathbf{x}}_0 e^{-\frac{1}{2}\int_0^t \beta(s)\mathrm{d}s}, \mathbf{I} - \mathbf{I}e^{-\int_0^t \beta(s)\mathrm{d}s})$$

The noise schedule $\beta(t) = at + b$ and $a, b$ are constants (usually $a = 0.1$ and $b = 19.9$), so that $\int_0^t \beta(s)\mathrm{d}s = at + 0.5bt^2$. The SNR at time $t$ is denoted as:

$$SNR(t) = \frac{e^{-\int_0^t \beta(s)\mathrm{d}s}}{1 - e^{-\int_0^t \beta(s)\mathrm{d}s}} = \frac{e^{-(at+0.5bt^2)}}{1 - e^{-(at+0.5bt^2)}} \tag{13}$$

The goal of SNR Scaling is to have the new $SNR'(t)$ such that

$$SNR'(t) = c \times SNR(t) \tag{14}$$

where $c \in \mathbb{R}$ is the introduced factor of SNR Scaling. Given $c$, we need to derive the new noise schedule $\beta'(t; c)$ for practical implementation. Let $y = \int_0^t \beta(s)\mathrm{d}s = at + 0.5bt^2$ and $y' = \int_0^t \beta'(s)\mathrm{d}s$, Eq. (14) becomes

$$c\frac{e^{-y}}{1 - e^{-y}} = \frac{e^{-y'}}{1 - e^{-y'}} \tag{15}$$

from which, we derive

$$e^{-y'} = c\frac{e^{-y}}{1 + (c-1)e^{-y}} \tag{16}$$

Taking the logarithm of both sides of Eq. (eq: SNR 4) yields

$$y' = y - \ln c + \ln[1 + (c-1)e^{-y}] \tag{17}$$

$$= at + 0.5bt^2 - \ln c + \ln[1 + (c-1)e^{-(at+0.5bt^2)}] \tag{18}$$

Take the derivative of Eq. (18) w.r.t $t$, we obtain the new noise schedule $\beta'(t)$

$$\beta'(t; c) = a + bt + \frac{(c-1)e^{-(at+0.5bt^2)}(-a - bt)}{1 + (c-1)e^{-(at+0.5bt^2)}} \tag{19}$$

In the code implementation, we apply Eq. (19) as the noise schedule to replace the original $\beta(t) = a + bt$. When the scaling factor $c = 1$, Eq. (19) degrades to $\beta(t)$.

**SNR Scaling for DPM-Solver.** In addition to applying $\beta'(t)$, we need to update the inverse function of $\lambda(t)$ defined in DPM-Solver (Lu et al., 2022) if we want to use it for sampling. The original $\lambda(t)$ is

$$\lambda(t) = 0.5\log(SNR(t)) \tag{20}$$

Now, given the updated $SNR'(t)$, we need to solve $t$ from $\lambda(t) = 0.5\log(SNR'(t))$.

$$\lambda = 0.5\log(SNR'(t)) \tag{21}$$

$$2\lambda = \log[\frac{e^{-y'}}{1 - e^{-y'}}] \tag{22}$$

$$e^{2\lambda} = \frac{e^{-y'}}{1 - e^{-y'}} \tag{23}$$

$$e^{-y'} = \frac{e^{2\lambda}}{1 + e^{2\lambda}} \tag{24}$$

$$y' = \ln[\frac{1 + e^{2\lambda}}{e^{2\lambda}}] \tag{25}$$

Plugging Eq. (18) into Eq. (25), we get

$$at + 0.5bt^2 - \ln c + \ln[1 + (c-1)e^{-(at+0.5bt^2)}] = \ln[\frac{1+e^{2\lambda}}{e^{2\lambda}}] \tag{26}$$

$$e^{at+0.5bt^2}[1 + (c-1)e^{-(at+0.5bt^2)}] = \frac{c(1+e^{2\lambda})}{e^{2\lambda}} \tag{27}$$

$$e^{at+0.5bt^2} + (c-1) = \frac{c(1+e^{2\lambda})}{e^{2\lambda}} \tag{28}$$

$$at + 0.5bt^2 = \ln[\frac{c(1+e^{2\lambda})}{e^{2\lambda}} + 1 - c] \tag{29}$$

$$t = \frac{-a + \sqrt{a^2 + 2b\ln[\frac{c(1+e^{2\lambda})}{e^{2\lambda}} + 1 - c]}}{b} \tag{30}$$

In practice, we apply Eq. (30) to update the inverse function of $\lambda(t)$. When the scaling factor $c = 1$, Eq. (30) degrades to the original inverse function $t = \frac{-a+\sqrt{a^2+2b\ln[\frac{1+e^{2\lambda}}{e^{2\lambda}}]}}{b}$.

### A.2.2. SNR SCALING IN DISCRETE-TIME DIFFUSION MODELS

Following DDPM (Ho et al., 2020) and DiT (Peebles & Xie, 2023), the forward perturbation kernel is

$$\mathcal{N}(\bar{\mathbf{x}}_t; \sqrt{\bar{\alpha}_t}\bar{\mathbf{x}}_0, (1 - \bar{\alpha}_t)\mathbf{I}) \tag{31}$$

where $\bar{\alpha}_t = \prod_{s=0}^{t} \alpha_s$ and $\alpha_t = 1 - \beta_t$. So that the original SNR at time $t$ is

$$SNR(t) = \frac{\bar{\alpha}_t}{1 - \bar{\alpha}_t} \tag{32}$$

From which, we obtain $\bar{\alpha}_t = \frac{SNR(t)}{SNR(t)+1}$. Now, scale the SNR by $c$, we have the updated signal schedule $\bar{\alpha}_t'$

$$\bar{\alpha}_t' = \frac{c \times SNR(t)}{c \times SNR(t) + 1} \tag{33}$$

Given the updated signal schedule $\bar{\alpha}_t'$, we could iteratively solve $\alpha_t'$ and $\beta_t'$, and use $\beta_t'$ for the implementation of SNR Scaling.

### A.3. Image Diffusion Is Spectral Autoregression

In this section, we discuss how the data information changes during the diffusion process of a diffusion model with the initial data being $\boldsymbol{x}_0$ and the data evolving into $\boldsymbol{x}_t$. For simplicity, we consider the scalar case of the diffusion process, since the original image diffusion is isotropic. Concretely, the forward diffusion SDE is

$$\mathrm{d}x_t = f(x_t, t)\mathrm{d}t + g(t)\mathrm{d}w_t, \tag{34}$$

where $t \in [0, T]$, $T > 0$ is a constant, $f(\cdot, \cdot)$ and $g(\cdot, \cdot)$ are the drift and diffusion coefficients respectively, and $w_t$ defines the standard Wiener process. We transform the image signal $x_t$ into $\hat{x}_t(\omega)$ using the Fourier transform. Here, $\omega$ represents the frequency. Thus, during the forward diffusion process, the signal $x_t$ can be represented in integral form as

$$x_t = x_0 + \int_0^t g(s)\mathrm{d}w_s \tag{35}$$

if $f(\cdot, \cdot) = 0$. After applying the Fourier transform, Eq. (35) becomes $\hat{x}_t(\omega) = \hat{x}_0(\omega) + \hat{\epsilon}_t(\omega)$ where $\epsilon_t(\omega)$ is the Fourier transform of the Gaussian noise term. Obviously, the mean value $\mathbb{E}[\hat{\epsilon}_t(\omega)] = 0$. We now prove that $\mathbb{E}\left[|\hat{\epsilon}_t(\omega)|^2\right] = \int_0^t |g(s)|^2 ds$. Consider the Fourier transform of $\epsilon_t(x)$:

$$\hat{\epsilon}_t(\omega) = \int_{-\infty}^{\infty} e^{-i\omega x} \epsilon_t(x) dx \tag{36}$$

Since $\epsilon_t(x)$ is a random process with zero mean, its Fourier transform $\hat{\epsilon}_t(\omega)$ is also a random variable with zero mean.

Calculate the variance of $\hat{\epsilon}_t(\omega)$:

$$\mathbb{E}[|\hat{\epsilon}_t(\omega)|^2] = \mathbb{E}\left[\left|\int_{-\infty}^{\infty} e^{-i\omega x}\epsilon_t(x)dx\right|^2\right] \tag{37}$$

Expand the square of the modulus:

$$\begin{aligned}
\mathbb{E}\left[|\hat{\epsilon}_t(\omega)|^2\right] &= \mathbb{E}\left[\int_{-\infty}^{\infty} e^{-i\omega x}\epsilon_t(x)\,dx \int_{-\infty}^{\infty} e^{i\omega y}\epsilon_t(y)\,dy\right] \\
&= \int_{-\infty}^{\infty}\int_{-\infty}^{\infty} e^{-i\omega(x-y)}\mathbb{E}\left[\epsilon_t(x)\epsilon_t(y)\right]\,dx\,dy.
\end{aligned} \tag{38}$$

Since $\epsilon_t(x)$ is spatially uncorrelated, we have

$$\mathbb{E}[\epsilon_t(x)\epsilon_t(y)] = \begin{cases} \int_0^t |g(s)|^2 ds, & \text{when } x = y \\ 0, & \text{when } x \neq y \end{cases} \tag{39}$$

This can be expressed as:

$$\mathbb{E}[\epsilon_t(x)\epsilon_t(y)] = \left(\int_0^t |g(s)|^2 ds\right)\delta(x-y). \tag{40}$$

where $\delta(.)$ is the Dirac delta function. Substitute Eq. (40) into Eq. (38):

$$\begin{aligned}
\mathbb{E}[|\hat{\epsilon}_t(\omega)|^2] &= \left(\int_0^t |g(s)|^2 ds\right)\int_{-\infty}^{\infty}\int_{-\infty}^{\infty} e^{-i\omega(x-y)}\delta(x-y)dxdy \\
&= \left(\int_0^t |g(s)|^2 ds\right)\int_{-\infty}^{\infty} e^{-i\omega(x-x)}dx \\
&= \left(\int_0^t |g(s)|^2 ds\right)\int_{-\infty}^{\infty} dx.
\end{aligned} \tag{41}$$

The integral $\int_{-\infty}^{\infty} dx$ means the integration region is infinite. In practice, we usually consider a finite spatial range or normalize the density. For simplicity, we consider a unit-length spatial range so that the integration result is 1, leading to

$$\mathbb{E}[|\hat{\epsilon}_t(\omega)|^2] = \int_0^t |g(s)|^2 ds \tag{42}$$

Then, we calculate the power density of various signals during the diffusion process. The power spectral density of signal $x_t$ is $S_{x_t}(\omega) = \mathbb{E}\left[|\hat{x}_t(\omega)|^2\right]$. After expansion, we obtain

$$\begin{aligned}
S_{x_t}(\omega) &= |\hat{x}_0(\omega)|^2 + 2\mathfrak{Re}\left(\hat{x}_0(\omega)\mathbb{E}\left[\hat{\epsilon}_t^*(\omega)\right]\right) + \mathbb{E}\left[|\hat{\epsilon}_t(\omega)|^2\right] \\
&= |\hat{x}_0(\omega)|^2 + 2\mathfrak{Re}\left(\hat{x}_0(\omega)\cdot 0\right) + \mathbb{E}\left[|\hat{\epsilon}_t(\omega)|^2\right] \\
&= |\hat{x}_0(\omega)|^2 + \mathbb{E}\left[|\hat{\epsilon}_t(\omega)|^2\right].
\end{aligned} \tag{43}$$

Since $\mathbb{E}\left[\hat{\epsilon}_t(\omega)\right] = 0$ and the cross term $[2\mathfrak{Re}\left(\hat{x}_0(\omega)\mathbb{E}\left[\hat{\epsilon}_t^*(\omega)\right]\right) = 0$, we have

$$S_{x_t}(\omega) = |\hat{x}_0(\omega)|^2 + \mathbb{E}\left[|\hat{\epsilon}_t(\omega)|^2\right] \tag{44}$$

where $\mathbb{E}\left[|\hat{\epsilon}_t(\omega)|^2\right] = \int_0^t |g(s)|^2 ds$, which completes the proof of Theorem 5.1.

Moreover, we can evaluate how much information is damaged during the forward diffusion process from the SNR perspective. A higher SNR implies that the signal is relatively purer and the degree of damage it undergoes is lower. On the contrary, a lower SNR indicates that the noise has a greater impact on the signal and the degree of information damage is also higher.

Consequently, the SNR ratio can intuitively reflect the noise level contaminating the information during the forward diffusion SDE. The SNR can be expressed as

$$\text{SNR}(\omega) = \frac{|\hat{\mathbf{x}}_0(\omega)|^2}{\mathbb{E}[|\hat{\boldsymbol{\epsilon}}_t(\omega)|^2]} = \frac{|\hat{\mathbf{x}}_0(\omega)|^2}{\int_0^t |g(s)|^2 ds} \tag{45}$$

We find that the change of SNR with frequency $\omega$ is completely determined by the power spectral density $|\mathbf{x}_0(\omega)|^2$ of the initial signal, while the noise power is the same at all frequencies.

For natural images, it is generally the case that they possess low-pass characteristics. Moreover, their power spectra typically conform to a power-law distribution (Turiel & Parga, 2000)), which can be expressed as $|\hat{x}_0(\omega)|^2 \propto |\omega|^{-\alpha}$, where $\alpha > 0$ denotes the spectral attenuation degree of the signal. Therefore, as the frequency $\omega$ increases, $|\hat{x}_0(\omega)|^2$ decreases rapidly. This indicates that $\text{SNR}(\omega)$ is low at high frequencies. As the diffusion time $t$ increases, the denominator $\int_0^t |g(s)|^2 ds$ increases, leading to an overall decrease in the SNR.

Given an SNR threshold $\gamma$ and a frequency $\omega$, the time $t_\gamma(\omega)$ when the SNR reaches the threshold $\gamma$ satisfies:

$$\text{SNR}(\omega) = \frac{|\omega|^{-\alpha}}{\int_0^{t_\gamma(\omega)} |g(s)|^2 ds} = \gamma \tag{46}$$

From this, we can solve $t_\gamma(\omega)$ to obtain the exact time required to reach $\text{SNR}(\omega) = \gamma$. The right-side image of Figure 5 provides the illustrations of Eq. (46).

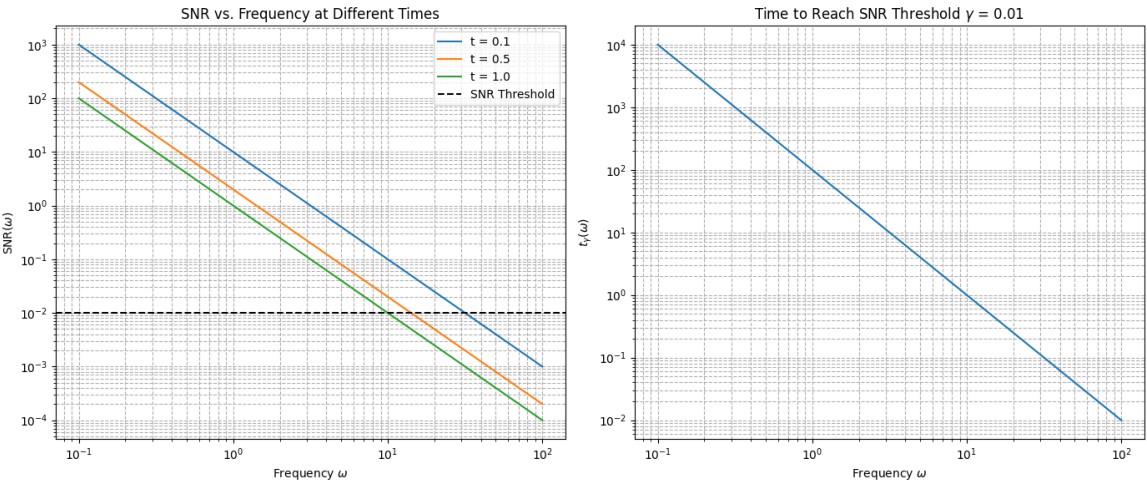

*Figure 5.* Relationship between frequency $\omega$ and SNR

## A.4. Averaged Power Spectral Density in DCT space

Dieleman (2024) uses the radially averaged power spectral density (RAPSD) to analyze the frequency of images in Fourier space. Similarly, given the diffusion perturbation kernel $\mathcal{N}(\boldsymbol{x}_t; \boldsymbol{x}_0 e^{-\frac{1}{2}\int_0^t \beta(s)ds}, \mathrm{I} - \mathrm{I}e^{-\int_0^t \beta(s)ds})$, we calculate the averaged power spectral density (APSD) for a clear image $\boldsymbol{x}_0$, the noisy image $\boldsymbol{x}_t$ and the isotropic Gaussian noise $\boldsymbol{\epsilon}_t$ ($\boldsymbol{\epsilon}_t = \sqrt{1 - e^{-\int_0^t \beta(s)ds}}\boldsymbol{\epsilon}$, where $\boldsymbol{\epsilon} \sim \mathcal{N}(\mathbf{0}, \boldsymbol{I})$). We use the Monte-Carlo method to estimate the APSD of $\boldsymbol{x}_0$, $\boldsymbol{x}_t$ and $\boldsymbol{\epsilon}_t$, respectively with 50,000 samples from the FFHQ 256×256 dataset. Figure 6 shows the APSD curves at different times. Similar to the RAPSD figures under the Fourier transform in Dieleman (2024), APSD also shows a pattern of frequency autoregression.

## A.5. Proof of DCT Upsampling Theorem

Consider a high-resolution image (e.g. $256 \times 256$) that consists of pixel blocks and each block is denoted as $\boldsymbol{A} \in \mathbb{R}^{2B \times 2B}$, then a low-resolution (e.g. $128 \times 128$) image is derived from the average pooling of the $256 \times 256$ image and each pixel

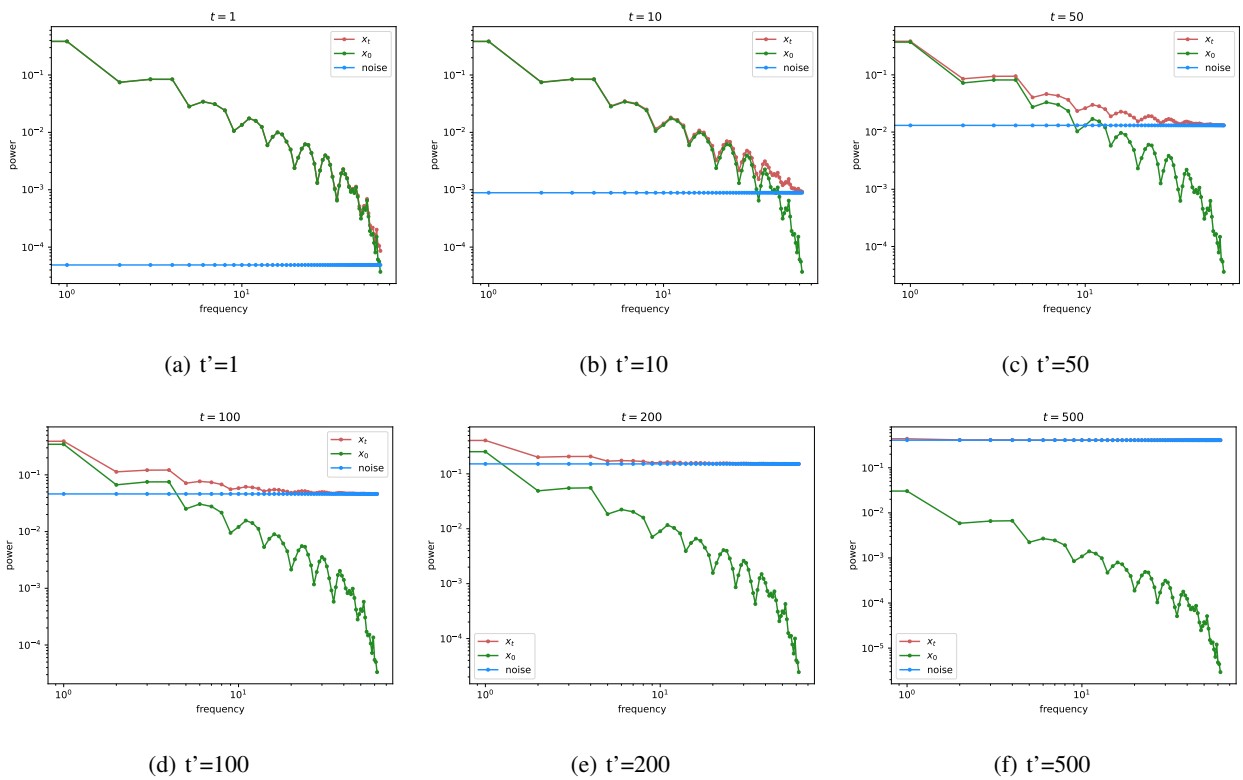

*Figure 6.* The averaged power spectral density (APSD) of $\boldsymbol{x}_0$, $\boldsymbol{x}_t$ and the noise $\boldsymbol{\epsilon}_t$ in the DCT space at time $t = t'/1000$.

block is denoted as $\bar{\boldsymbol{A}} \in \mathbb{R}^{B \times B}$. In this case, these two images have the same number of DCT blocks. Let $\boldsymbol{D} \in \mathbb{R}^{2B \times 2B}$ be the DCT block converted from $\boldsymbol{A} \in \mathbb{R}^{2B \times 2B}$ and $\bar{\boldsymbol{D}} \in \mathbb{R}^{B \times B}$ be the DCT block converted from $\bar{\boldsymbol{A}} \in \mathbb{R}^{B \times B}$. According to Eq. (5), we have

$$D(u, v) = \sqrt{\frac{2}{2B}} \sqrt{\frac{2}{2B}} \sum_{x=0}^{2B-1} \sum_{y=0}^{2B-1} A(x, y) \cos \left[ \frac{(2x+1)u\pi}{4B} \right] \cos \left[ \frac{(2y+1)v\pi}{4B} \right] \tag{47}$$

$$\bar{D}(u, v) = \sqrt{\frac{2}{B}} \sqrt{\frac{2}{B}} \sum_{i=0}^{B-1} \sum_{j=0}^{B-1} \bar{A}(i, j) \cos \left[ \frac{(2i+1)u\pi}{2B} \right] \cos \left[ \frac{(2j+1)v\pi}{2B} \right] \tag{48}$$

where $D(u, v)$ is an element of $\boldsymbol{D}$ and $\bar{D}(u, v)$ is an element of $\bar{\boldsymbol{D}}$, respectively. Since $\bar{\boldsymbol{A}} \in \mathbb{R}^{B \times B}$ is the average pooling of $\boldsymbol{A} \in \mathbb{R}^{2B \times 2B}$, we have

$$\bar{A}(i, j) = \frac{1}{4} \sum_{m=0}^{1} \sum_{n=0}^{1} A(2i+m, 2j+n). \tag{49}$$

Plug Eq. (49) into Eq. (48), we obtain

$$\bar{D}(u, v) = \sqrt{\frac{2}{B}} \sqrt{\frac{2}{B}} \sum_{i=0}^{B-1} \sum_{j=0}^{B-1} \frac{1}{4} \sum_{m=0}^{1} \sum_{n=0}^{1} A(2i+m, 2j+n) \cos \left[ \frac{(2i+1)u\pi}{2B} \right] \cos \left[ \frac{(2j+1)v\pi}{2B} \right] \tag{50}$$

Apply change of variable $x = 2i + m$ and $y = 2j + n$, Eq. (50) becomes

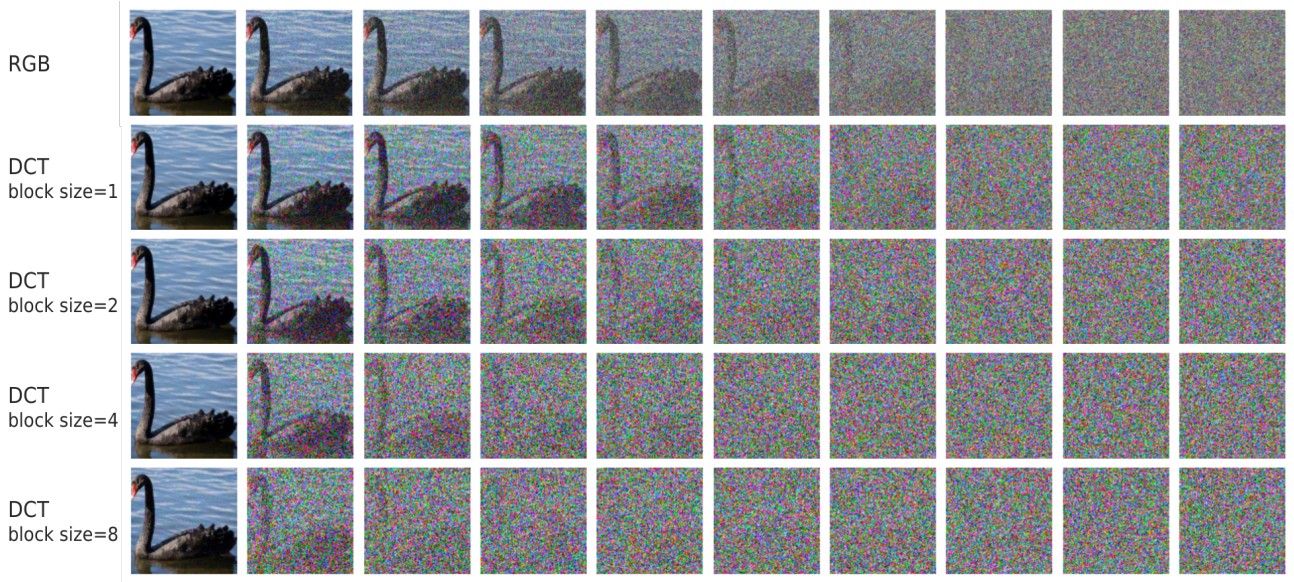

*Figure 7.* The forward perturbation process of RGB and DCT using the same forward SDE $\mathrm{d}\boldsymbol{x}_t = \boldsymbol{f}(\boldsymbol{x}_t, t)\mathrm{d}t + g(t)\mathrm{d}\boldsymbol{w}_t$.

$$\bar{D}(u,v) = \frac{1}{2B} \sum_{x=0}^{2B-1} \sum_{y=0}^{2B-1} A(x,y) \sum_{m=0}^{1} \cos\left[\frac{(x-m+1)u\pi}{2B}\right] \sum_{n=0}^{1} \cos\left[\frac{(y-n+1)v\pi}{2B}\right] \tag{51}$$

$$= \frac{1}{2B} \sum_{x=0}^{2B-1} \sum_{y=0}^{2B-1} A(x,y) \left( \underbrace{\cos\left[\frac{(x+1)u\pi}{2B}\right]}_{m=0} + \underbrace{\cos\left[\frac{xu\pi}{2B}\right]}_{m=1} \right) \left( \underbrace{\cos\left[\frac{(y+1)v\pi}{2B}\right]}_{n=0} + \underbrace{\cos\left[\frac{yv\pi}{2B}\right]}_{n=1} \right) \tag{52}$$

However, since $i = \frac{x-m}{2}$ is an integer when applying the change of variable, $x$ and $m$ must both be odd or both be even (the same applies to $y$ and $n$). Therefore, for any $x, y$ in Eq. (52), only one term exists within each of the two big parentheses. An approximation can be obtained by taking the average of the two cosine terms within each bracket, which gives

$$\bar{D}(u,v) \approx \frac{1}{2B} \sum_{x=0}^{2B-1} \sum_{y=0}^{2B-1} A(x,y) \frac{1}{2} \left( \cos\left[\frac{(x+1)u\pi}{2B}\right] + \cos\left[\frac{xu\pi}{2B}\right] \right) \frac{1}{2} \left( \cos\left[\frac{(y+1)v\pi}{2B}\right] + \cos\left[\frac{yv\pi}{2B}\right] \right) \tag{53}$$

$$= \frac{1}{2B} \sum_{x=0}^{2B-1} \sum_{y=0}^{2B-1} A(x,y) \left( \cos\left[\frac{(2x+1)u\pi}{4B}\right] \cos\left[\frac{u\pi}{4B}\right] \right) \left( \cos\left[\frac{(2y+1)v\pi}{4B}\right] + \cos\left[\frac{v\pi}{4B}\right] \right) \tag{54}$$

$$= \frac{1}{2B} \cos\left[\frac{u\pi}{4B}\right] \cos\left[\frac{v\pi}{4B}\right] \sum_{x=0}^{2B-1} \sum_{y=0}^{2B-1} A(x,y) \cos\left[\frac{(2x+1)u\pi}{4B}\right] \cos\left[\frac{(2y+1)v\pi}{4B}\right] \tag{55}$$

From Eq. (53) to Eq. (54) we apply the trigonometric formulas

$$\cos(A) + \cos(B) = 2\cos(\frac{A+B}{2})\cos(\frac{A-B}{2})$$

Now comparing Eq. (55) with Eq. (47), we obtain

$$\bar{D}(u,v) \approx \frac{1}{2} \cos\left[\frac{u\pi}{4B}\right] \cos\left[\frac{v\pi}{4B}\right] D(u,v) \tag{56}$$

which completes the proof of Theorem 5.2.

### A.6. Training Parameters of UViT, DiT and DCTdiff

We list the model and training parameters in Table 9 and Table 10 where the former compares UViT and DCTdiff (inherited from UViT) and the latter compares DiT and DCTdiff (inherited from DiT). We use the default training settings from UViT and DiT without any change.

Regarding the choice of DCTdiff parameters, we find that the fixed $\tau = 98$ used for Entropy-Consistent Scaling is effective on all datasets, possibly due to the statistical consistency of image frequency distributions. The block size $B$ and SNR Scaling factor $c$ only depend on the image resolution, one can refer to Table 9 to determine $B$ and $c$ given a new dataset. Finally, the frequency elimination parameter $m^*$ can be calculated from Eq. (6).

*Table 9.* Training and network parameters of UViT and DCTdiff on different datasets.

| Dataset | Model | Transformer parameters | | | Learning parameters | | DCTdiff parameters | | |
|---|---|---|---|---|---|---|---|---|---|
| | | # parameters | patch size | # tokens | batch size | learning rate | $\tau$ | $m^*$ | $c$ |
| CIFAR-10 | UViT | 131M | 4 | 64 | 256 | 0.0002 | - | - | - |
| | DCTdiff | 131M | 4 ($B = 2$) | 64 | 256 | 0.0002 | 98.25 | 0 | 4 |
| CelebA 64 | UViT | 44M | 4 | 256 | 256 | 0.0002 | - | - | - |
| | DCTdiff | 44M | 4 ($B = 2$) | 256 | 256 | 0.0002 | 98.25 | 0 | 4 |
| ImageNet 64 | UViT | 44M | 4 | 256 | 1024 | 0.0003 | - | - | - |
| | DCTdiff | 44M | 4 ($B = 2$) | 256 | 1024 | 0.0003 | 98.25 | 0 | 4 |
| FFHQ 128 | UViT | 44M | 8 | 256 | 256 | 0.0002 | - | - | - |
| | DCTdiff | 44M | 8 ($B = 4$) | 256 | 256 | 0.0002 | 98.25 | 7 | 4 |
| FFHQ 256 | UViT (latent) | 131M + 84M | 2 | 256 | 256 | 0.0002 | - | - | - |
| | DCTdiff | 131M | 8 ($B = 4$) | 1024 | 256 | 0.0002 | 98.25 | 8 | 4 |
| FFHQ 512 | UViT (latent) | 131M + 84M | 4 | 256 | 128 | 0.0001 | - | - | - |
| | DCTdiff | 131M | 16 ($B = 8$) | 1024 | 128 | 0.0001 | 98.25 | 46 | 12 |
| AFHQ 512 | UViT (latent) | 131M + 84M | 4 | 256 | 128 | 0.0001 | - | - | - |
| | DCTdiff | 131M | 16 ($B = 8$) | 1024 | 128 | 0.0001 | 98.25 | 46 | 12 |

*Table 10.* Training and network parameters of DiT and DCTdiff on different datasets.

| Dataset | Model | Transformer parameters | | | Learning parameters | | DCTdiff parameters | | |
|---|---|---|---|---|---|---|---|---|---|
| | | # parameters | patch size | # tokens | batch size | learning rate | $\tau$ | $m^*$ | $c$ |
| CelebA 64 | DiT | 58M | 4 | 256 | 256 | 0.0001 | - | - | - |
| | DCTdiff | 58M | 4 ($B = 2$) | 256 | 256 | 0.0001 | 98.25 | 0 | 4 |
| FFHQ 128 | DiT | 58M | 8 | 256 | 256 | 0.0001 | - | - | - |
| | DCTdiff | 58M | 7 ($B = 4$) | 256 | 256 | 0.0001 | 98.25 | 8 | 4 |

### A.7. Evaluation of IS, Precision, Recall, and CMMD

In addition to the FID reported in the main paper, our evaluation of UViT and DCTdiff also includes Inception Score (IS) (Salimans et al., 2016), Recall, Precision (Kynkäänniemi et al., 2019), and CLIP Maximum Mean Discrepancy (CMMD) (Jayasumana et al., 2024). As shown in Table 11, DCTdiff achieves better IS, CMMD, and mostly exhibits higher Precision and Recall than UViT.

*Table 11.* Comparison between UViT and DCTdiff on CIFAR-10, FFHQ 128 and AFHQ 512. We use DDIM sampler (NFE=100) for CIFAR-10 and DPM-Solver (NFE=100) for the other two datasets.

| Model | CIFAR-10 | | | | FFHQ 128 | | | | AFHQ 512 | | | |
|---|---|---|---|---|---|---|---|---|---|---|---|---|
| | CMMD ↓ | IS ↑ | Precision ↑ | Recall ↑ | CMMD ↓ | IS ↑ | Precision ↑ | Recall ↑ | CMMD ↓ | IS ↑ | Precision ↑ | Recall ↑ |
| UViT | 0.052 | 7.08 | **0.668** | 0.589 | 0.610 | 3.54 | 0.648 | 0.485 | 0.373 | 11.00 | 0.547 | 0.496 |
| DCTdiff | **0.043** | **7.70** | 0.660 | **0.606** | **0.470** | **3.67** | **0.668** | **0.512** | **0.335** | 11.00 | **0.632** | 0.496 |

## A.8. Scalability of DCTdiff on ImageNet 64×64

Similar to the scalability experiments on FFHQ 128, we apply the network scaling strategy 'small, mid, and mid deep' on ImageNet 64 to further test the scalability of DCTdiff. Using the DDIM sampler and NFE= 100, we summarize the FID-50k results:

- DCTdiff (small), FID=8.69.

- DCTdiff (mid), FID=4.67.

- DCTdiff (mid, deep), FID=4.30.

- FID comparison: 4.67 (DCTdiff, mid) vs. 5.85 (UViT, mid)

## A.9. Inference Time and Generation Quality

In Section 6.3, we report the wall-clock inference time of DCTdiff and UViT without considering their sampling quality. However, when considering inference time at comparable generation quality, DCTdiff shows clear advantages: latent UViT requires 20 minutes to achieve FID=10.89, whereas DCTdiff achieves FID=8.04 in just 9 minutes on FFHQ 512.

*Table 12.* Wall-clock inference time on FFHQ 512×512. FID-50k is reported using DPM-Solver

| Model | NFE | | | |
|---|---|---|---|---|
| | 100 | 50 | 20 | 10 |
| UViT (latent) | **20.2 min (FID 10.89)** | 13.4 min (FID 10.94) | 9.3 min (FID 11.31) | 7.9 min (FID 23.61) |
| DCTdiff | 47.8 min (FID 7.07) | 23.9 min (FID 7.09) | **9.6 min (FID 8.04)** | 4.8 min (FID 19.67) |

*Table 13.* Wall-clock inference time on AFHQ 512×512. FID-50k is reported using DPM-Solver

| Model | NFE | | | |
|---|---|---|---|---|
| | 100 | 50 | 20 | 10 |
| UViT (latent) | **20.2 min (FID 10.86)** | 13.4 min (FID 10.86) | 9.3 min (FID 11.94) | 7.9 min (FID 28.31) |
| DCTdiff | 47.8 min (FID 8.76) | 23.9 min (FID 8.87) | **9.6 min (FID 10.05)** | 4.8 min (FID 21.05) |

## A.10. Application of DCT Upsampling in Super-resolution Image Generation

ADM (Dhariwal & Nichol, 2021) proposed a neural network-based upsampling method for super-resolution image generation. In detail, a diffusion model $D_l$ generates a low-resolution (e.g., 64×64) image. This image is then upsampled to the target resolution (e.g., 256×256) using pixel interpolation. The upsampled image, as a condition, is finally fed into another diffusion model $D_h$ to learn the data distribution of the high-resolution images. Following this pipeline, we replace the pixel interpolation with DCT Upsampling to verify its feasibility in super-resolution generation. We evaluate the FID-10K using DDPM sampler (NFE=50) on ImageNet 256×256. Using the same pretrained model $D_h$, DCT Upsampling achieves FID 14.53, which is inferior to the pixel interpolation (FID 10.71). We believe this phenomenon is caused by the training bias, where $D_h$ was only trained on the pixel interpolation images and had never seen the DCT upsampled images. A fair comparison is that $D_h$ should be trained on the DCT upsampled images. We leave this to future work.

# B. Ablation Study

In this section, we elaborate on the effect of each design factor of DCTdiff using the dataset FFHQ 128×128 and the base model UViT. We report the FID-10k using DPM-solver throughout the ablation section. Table 14 presents the ablation results and the first row shows FID-10k achieved by UViT during training. We then progressively added each design element to the previous base model to examine the design space of DCTdiff:

- UViT (YCbCr) inherits from UViT and replaces the RGB pixel inputs with YCbCr inputs.

- DCTdiff (ECS, $m = 0$) integrates the DCT transformation and Entropy-Consistent Scaling into UViT (YCbCr).

- DCTdiff (ECS, $m = 7$) further eliminates 7 high-frequency coefficients.

- DCTdiff (EBFR) adds Entropy-Based Frequency Reweighting on DCTdiff (ECS, $m = 7$).

- DCTdiff (SNR) incorporates the SNR Scaling based on DCTdiff (EBFR)

*Table 14.* Ablation study of DCTdiff design factors: FID-10k of UViT and DCTdiff during training on FFHQ 128×128. The convergence FID is marked as bold at each row.

| Row | Model | Training steps | | | | | | | | | | | | | |
|-----|-------|------|------|------|------|------|------|------|------|------|------|------|------|------|------|
| | | 100k | 150k | 200k | 250k | 300k | 350k | 400k | 450k | 500k | 550k | 600k | 650k | 700k | 750k |
| 1 | UViT | 70.67 | 40.64 | 24.72 | 17.88 | 14.73 | 13.65 | 12.64 | 12.10 | 11.27 | 11.17 | 11.02 | 10.84 | **10.58** | 10.60 |
| 2 | UViT (YCbCr) | 17.70 | **14.79** | 16.38 | - | - | - | - | - | - | - | - | - | - | - |
| 3 | DCTdiff (ECS, $m = 0$) | 26.59 | 17.11 | 16.90 | 16.29 | 14.38 | 13.42 | **12.63** | 12.92 | - | - | - | - | - | - |
| 4 | DCTdiff (ECS, $m = 8$) | 14.11 | 12.42 | 10.81 | 10.52 | 10.24 | **9.74** | 9.76 | - | - | - | - | - | - | - |
| 5 | DCTdiff (ECS, $m = 7$) | 36.98 | 14.79 | 11.14 | 10.65 | **9.50** | 9.68 | - | - | - | - | - | - | - | - |
| 6 | DCTdiff (EBFR) | 15.66 | 10.61 | 9.69 | 9.72 | **9.09** | 9.20 | - | - | - | - | - | - | - | - |
| 7 | DCTdiff (SNR, $c = 2$) | 18.32 | 9.19 | 8.54 | 8.32 | **7.94** | 9.16 | - | - | - | - | - | - | - | - |
| 8 | DCTdiff (SNR, $c = 4$) | 20.60 | 18.90 | 8.49 | 8.07 | **7.64** | 7.73 | - | - | - | - | - | - | - | - |

## B.1. Ablation Study: YCbCr Accelerates the Diffusion Training

To evaluate the effect of YCbCr color space transformation in DCTdiff, we substitute the RGB inputs with YCbCr (2x chroma subsampling) input. The corresponding results are shown in the second row of Table 14, indicating that YCbCr with chroma subsampling dramatically accelerates the diffusion training but at the cost of generative quality (FID-10k increases from 10.58 to 14.79). We believe the chroma subsampling provides the training acceleration, but the reduced color redundancy causes a drop in generation quality.

## B.2. Ablation Study: DCT and Entropy-Consistent Scaling

As we mentioned in Section 4.4, Entropy-Consistent Scaling (ECS) is a key factor making the DCT generative modeling effective. In detail, DCTdiff (ECS) not only enjoys the training acceleration benefit of YCbCr subsampling, but also yields a better generative quality than UViT (YCbCr) (see Table 14). We attribute the improvement of generation quality to the DCT space where low-frequency coefficients occupy the majority of image information.

## B.3. Ablation Study: Eliminating $m$ High-frequency Coefficients

In Table 14, we show the effect of eliminating $m$ high-frequency coefficients in each DCT block. The block size $B$ is 4 and $m = 0$ refers to maintaining all coefficients for diffusion training and sampling. By comparing Row 3, Row 4 and Row 5 of Table 14, it is clear that ignoring a suitable amount of high-frequency signals increases the generative quality and boosts the training, too. The optimal $m$ can be decided via Eq. (6).

## B.4. Ablation Study: Entropy-Based Frequency Reweighting

In Section 5.1, we highlight the frequency prioritization property of DCT image modeling in which some frequency coefficients can be modeled preferentially according to the task prior knowledge. We adopt the Entropy-Based Frequency Reweighting (EBFR) for image generative modeling tasks as low-frequency coefficients have large entropy and contribute more to the visual quality of images than high-frequency signals. Row 5 and Row 6 of Table 14 demonstrate that EBFR improves the generation quality of DCTdiff without affecting the training convergence.

## B.5. Ablation Study: SNR Scaling $c$ of Noise Schedule

Since the block size $B$ affects the forward perturbation process of DCTdiff (detailed in Section 4.5), we propose SNR Scaling for DCTdiff to scale the noise schedule of UViT by a constant $c$. Row 7 and Row 8 of Table 14 show that SNR Scaling significantly improves the generative quality of DCTdiff and a wide range of parameter $c$ can yield the FID improvement. We also visualize the effect of $c$ in the perturbation process of DCTdiff in Figure 8 where the image size is 128×128 and the block size is 4.

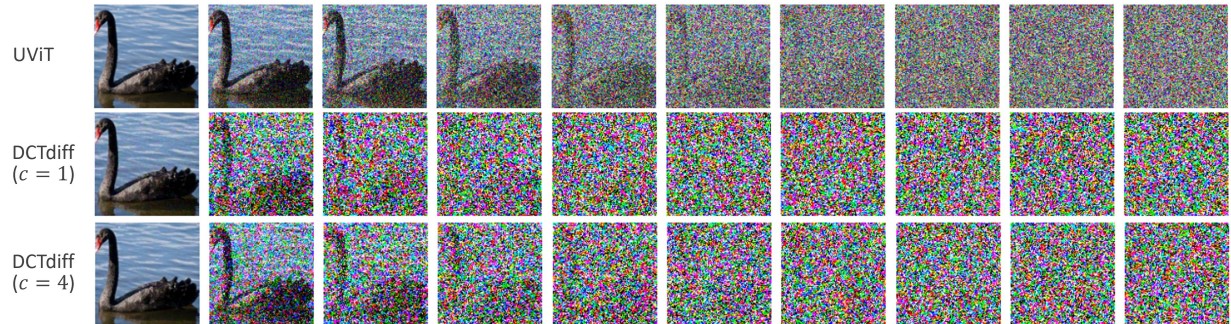

*Figure 8.* Visualization of the forward SDE process. UViT and DCTdiff ($c = 1$) share the same noise schedule, while DCTdiff ($c = 4$) scales up the noise schedule by a factor of 4.

### B.6. Ablation Study: Block Size $B$

Although we have provided the choice of block size $B$ for image resolutions ranging from 32×32 to 512×512 in Table 9, it is worth discussing its effect and importance to diffusion modeling in detail. Similar to the conclusion of patch size $P$ mentioned in UViT (Bao et al., 2023) and DiT (Peebles & Xie, 2023), we also confirm that the block size $B$ used in diffusion Transformers must be relatively small (smaller than the usual $P = 16$ in image classification). Since we always have the relationship $P = 2B$, one can determine $B$ based on the value of $P$ used in UViT and DiT when the resolution is below 256×256. Additionally, the value of $B$ greatly affects the generative quality and training convergence of DCTdiff: a larger $B$ leads to faster training but sacrifices the generation quality. In Table 15, we show the FID of DCTdiff on the dataset FFHQ 256×256 with different $B$.

*Table 15.* Ablation study of block size $B$: FID-10k of DCTdiff during training on FFHQ 256×256 with different block sizes. The convergence FID is marked as bold in each row.

| Model | Training steps | | | | | | |
|---|---|---|---|---|---|---|---|
| | 50k | 100k | 150k | 200k | 250k | 300k | 350k |
| DCTdiff (B=4) | 125.76 | 44.67 | 10.58 | 8.94 | 7.07 | **6.97** | 7.39 |
| DCTdiff (B=8) | 213.14 | 19.75 | **18.41** | 20.37 | - | - | - |

## C. Qualitative Results

### C.1. Qualitative Comparison between VAE Compression and DCT Compression

We randomly sample several images from ImageNet 256×256 dataset, then we perform VAE compression and DCT compression (4× compression ratio). The reconstructed images after compression are shown in Figure 9. From this, we clearly see that VAE compression loses image details and local image structure while DCT compression maintains most of the image information. Also, we find that VAE compression is not good at reconstructing letters, digital numbers, and unseen images (not trained by VAE). In contrast, DCT compression is training-free, computationally negligible, and insensitive to image domains.

### C.2. Qualitative Results of DCT Upsampling

In Figure 10, we show the visual differences between Pixel Upsampling by bicubic interpolation and DCT Upsampling.

### C.3. Qualitative Results of DCTdiff

We show the uncurated samples generated by DCTdiff:

- Figure 11, ImageNet 64×64, FID=4.30, generated by DDIM sampler with NFE=100

- Figure 12, FFHQ 128×128, FID=4.98, generated by DPM-Solver with NFE= 100

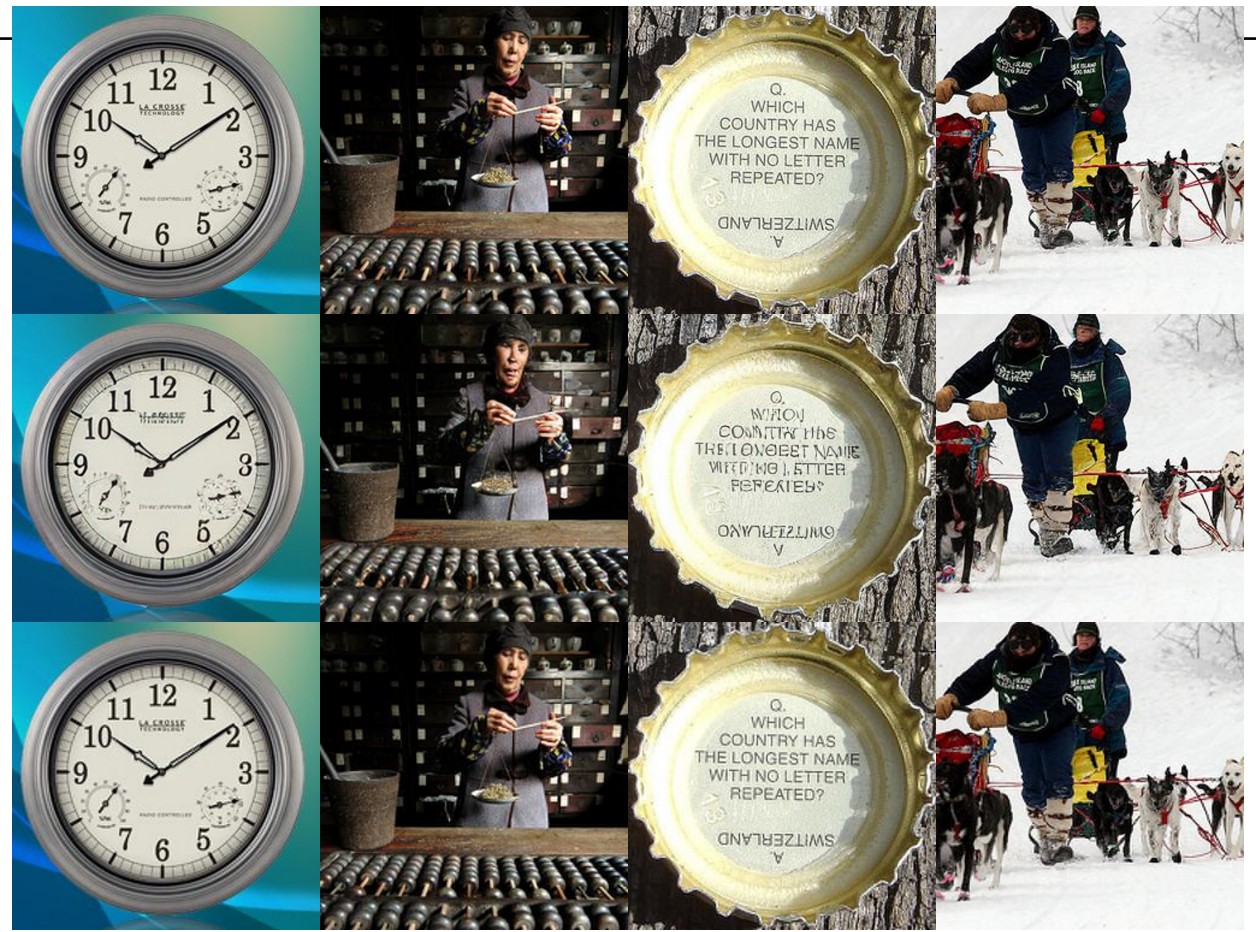

*Figure 9.* Qualitative comparison between VAE compression and DCT compression. The first row shows the raw images (sampled from ImageNet 256×256). The second and third rows present the reconstructed images after VAE compression and DCT compression (4×).

- Figure 13, FFHQ 256×256, FID=5.08, generated by DPM-Solver with NFE= 100

- Figure 14, FFHQ 512×512, FID=7.07, generated by DPM-Solver with NFE= 100

- Figure 15, AFHQ 512×512, FID=8.76, generated by DPM-Solver with NFE= 100

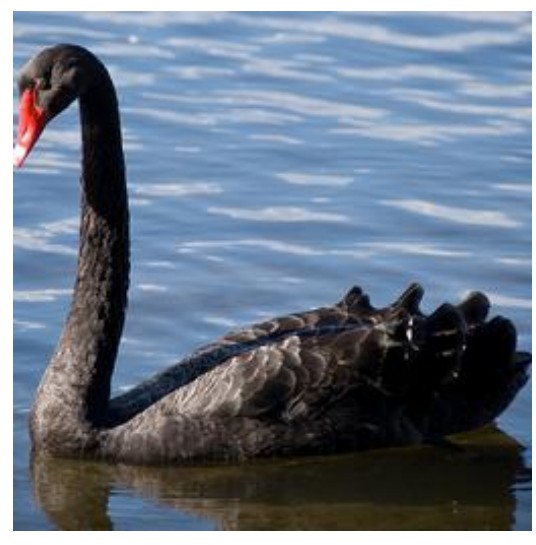

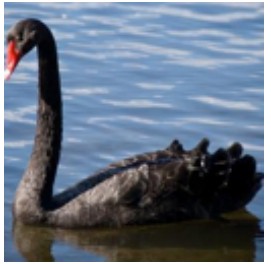

(b) 128×128 (downsampled from 256×256)

(a) 256×256 (ground truth)

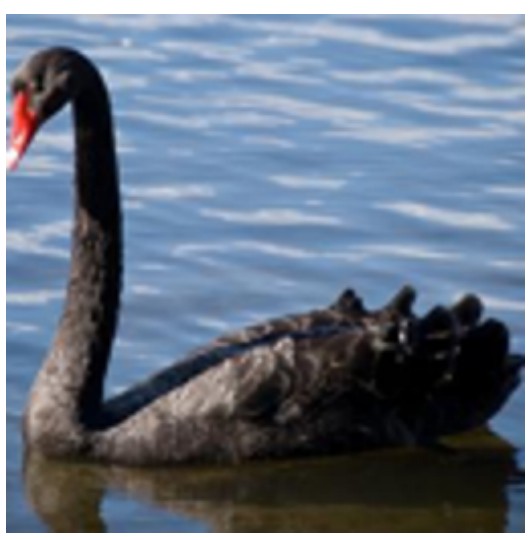

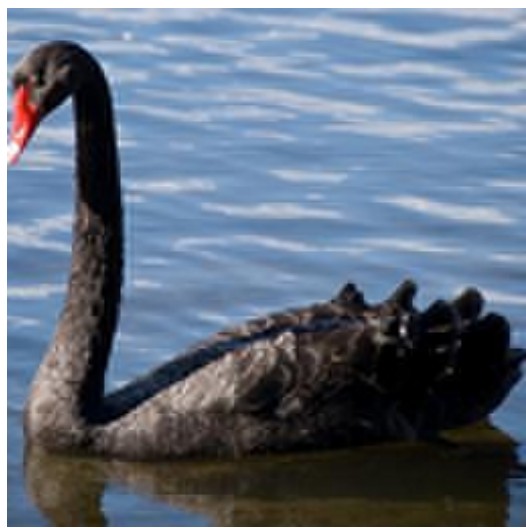

(c) 256×256 (Pixel Upsampling)

(d) 256×256 (DCT Upsampling)

*Figure 10.* Comparison between Pixel Upsampling and our proposed DCT Upsampling. 10(c) and 10(d) are upsampled based on 10(b).

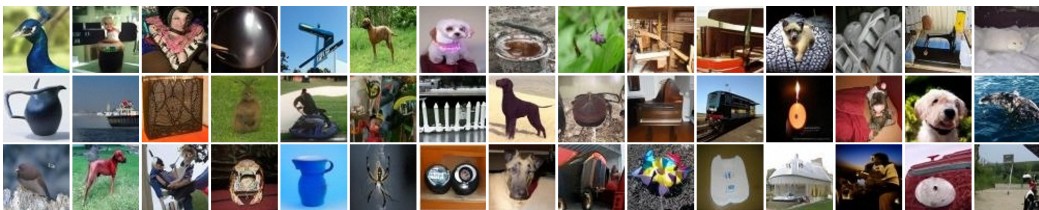

*Figure 11.* Uncurated samples generated by DCTdiff trained on the dataset ImageNet 64×64 (FID= 4.30).

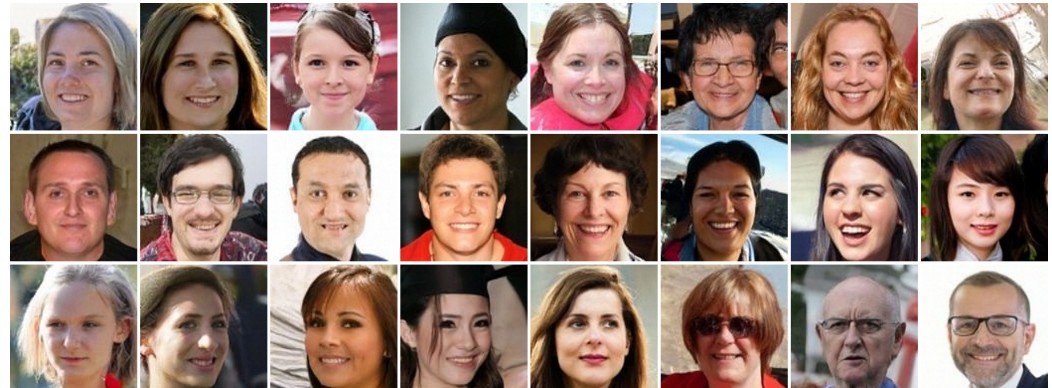

*Figure 12.* Uncurated samples generated by DCTdiff trained on the dataset FFHQ 128×128 (FID= 4.98).

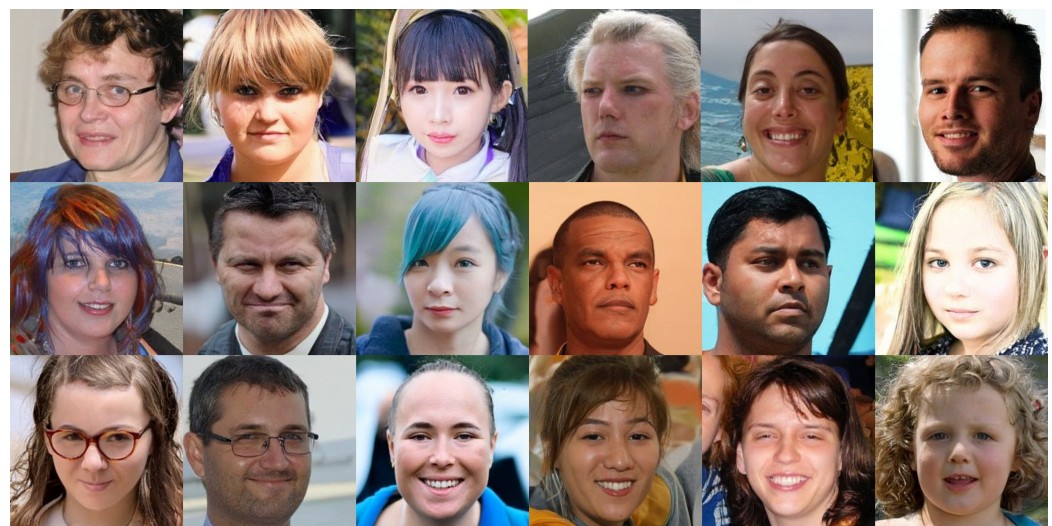

*Figure 13.* Uncurated samples generated by DCTdiff trained on the dataset FFHQ 256×256 (FID= 5.08).

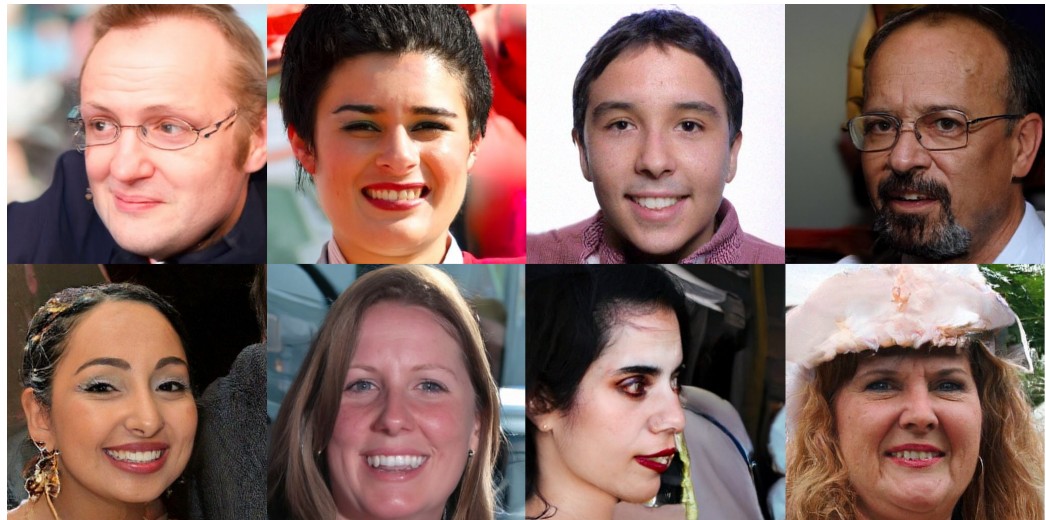

*Figure 14.* Uncurated samples generated by DCTdiff trained on the dataset FFHQ 512×512 (FID= 7.07).

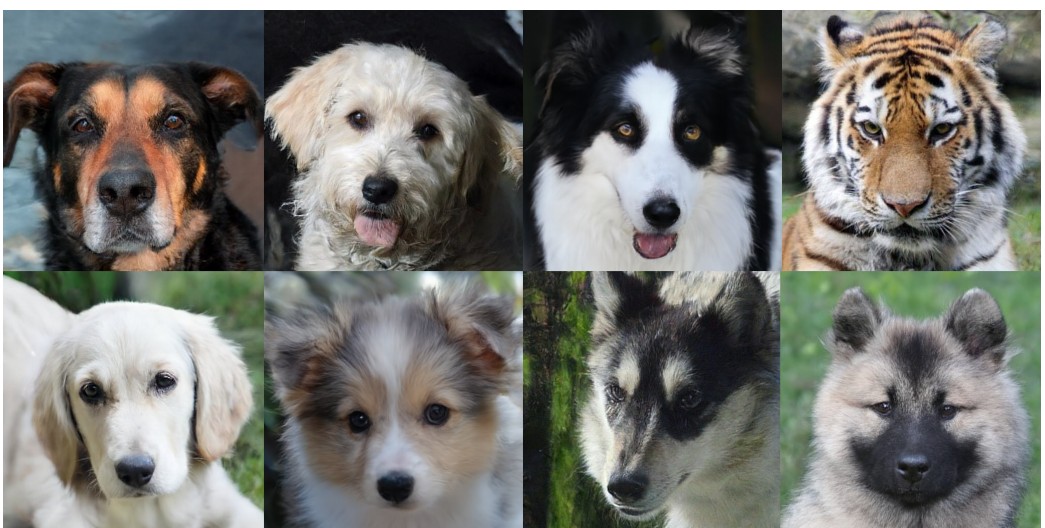

*Figure 15.* Uncurated samples generated by DCTdiff trained on the dataset AFHQ 512×512 (FID= 8.76).

