# OpenReview forum: "DCTdiff: Intriguing Properties of Image Generative Modeling in the DCT Space"
_ICML.cc/2025/Conference — ICML 2025 poster_

### Official Review · Reviewer_guDg · 2025-03-11

**Overall Recommendation:** 4

**Summary:**

This paper introduces a new paradigm in diffusion models by using DCT coefficients, specifically low-frequency components, as operands instead of pixel or latent representations. Inspired by JPEG compression, this method aims to improve efficiency. The model achieves 512x512 resolution without latent representations. The paper also claims that diffusion models can be interpreted as spectral autoregression.

**Claims And Evidence:**

The core idea is interesting, which is the primary reason why I recommend this paper, but some claims lack convincing theoretical or experimental support.

**Lack of qualitative evidence**: Unlike related works such as VAR (Tian et al., 2024) and LCM (Luo et al., 2023), this paper does not provide enough qualitative results in the main text. Typically, such studies include extensive uncurated results to prevent cherry-picking concerns. However, the main paper lacks qualitative figures, and the appendix only includes Figs. 9-11, missing key results for FFHQ 512.

**Overclaim on spectral autoregression**: Unlike VAR, which is truly autoregressive in nature, the proposed model is not structurally autoregressive. Diffusion models naturally generate low-frequency components early and high-frequency components later, a well-known phenomenon (Patashnik et al., ICCV 2023). This claim lacks novelty and does not provide practical benefits for DCTdiff.

**Essential References Not Discussed:**

The claim that "diffusion models are spectral autoregression" is not novel. Unlike true autoregressive models like VAR, diffusion models do not have structural autoregression. They just *tend* to make low-frequency in the early stages, and then *tend* to refine high-frequency in the later stages. The authors should cite Or Patashnik et al. (ICCV 2023) and tone down this claim.

### ref:
Patashnik, Or, et al. "Localizing object-level shape variations with text-to-image diffusion models." Proceedings of the IEEE/CVF international conference on computer vision. 2023.

**Experimental Designs Or Analyses:**

The main experiments and analyses are generally sound.

**Methods And Evaluation Criteria:**

The core idea of using DCT coefficients as diffusion model inputs is strong. However, evaluation has weaknesses.
The metrics and baselines are limited (only FID and UViT), which may not ensure a fair comparison. UViT was chosen for parameter consistency, but there is no clear reason why other pretrained models were not evaluated. A broader evaluation with models trained on the same datasets (even if #Parameters, GFLOPs, and training steps differ) would improve the study. Additional metrics, such as Inception Score, could be included, but more qualitative results would be the most impactful improvement.

**Other Comments Or Suggestions:**

The authors should provide more qualitative results, and it would be better that they go to the first figure.

**Other Strengths And Weaknesses:**

The study tackles an important problem, and its ideas could extend to many future works, such as Video diffusion models. This makes it a promising research direction.

**Questions For Authors:**

1. Why are qualitative results so limited in the main paper and appendix? Can you provide more uncurated samples?

2. Why were only UViT-based baselines used? Can you compare DCTdiff against more diverse models, even if their training settings differ?

**Relation To Broader Scientific Literature:**

The paper mainly compares DCTdiff with UViT-based latent diffusion using FID scores. However, the advantages of DCTdiff may extend beyond what is discussed.

For example, latent diffusion models using VAEs typically produce high-quality images but introduce issues with encoder-decoder invertibility. Hong et al., NeurIPS 2024, propose a correction algorithm for this, suggesting that DCTdiff could bypass such issues entirely.

### refs:
Hong, Seongmin, et al. "Gradient-free Decoder Inversion in Latent Diffusion Models." Advances in Neural Information Processing Systems 37 (2024): 82982-83007.

**Theoretical Claims:**

Theorems 5.1 and 5.2 appear correct but do not significantly contribute to the main claims.

Theorem 5.1: States that diffusion removes high-frequency components first and low-frequency components later. This is well known and not specific to DCTdiff.

Theorem 5.2: Reformulates existing DCT-based upsampling methods in a mathematical form. This principle is already widely used and not novel.

Overall, these theorems do not offer strong contributions.

---

> ### Author Rebuttal · Authors · 2025-04-01
>
> Thank you very much for the reviews and constructive suggestions.
>
> **Q1: the main paper lacks qualitative figures, and the appendix only includes Figs. 9-11, missing key results for FFHQ 512**
>
> We will add more qualitative samples in the appendix, including the randomly drawn ones, and compare them with the baseline as well. Surely, the qualitative results on FFHQ 512x512 will be added to the paper, too.
>
> **Q2: Theorems 5.1 and 5.2 appear correct but do not significantly contribute to the main claims. Theorem 5.1: States that diffusion removes high-frequency components first and low-frequency components later. This is well known (Patashnik et al., ICCV 2023) and not specific to DCTdiff. Theorem 5.2: Reformulates existing DCT-based upsampling methods in a mathematical form.**
>
> Theorem 5.1 provides a theoretical foundation for understanding how noise affects different frequency components in the diffusion process. It formalizes the observation that high-frequency signals are more susceptible to noise perturbations due to the energy concentration properties of the DCT. This insight is crucial for justifying why DCTdiff exhibits a faster noise accumulation process than pixel diffusion, ultimately motivating the introduction of the SNR scaling method. Regarding Theorem 5.2, we present an alternative derivation that differs from the existing approach in the literature. We believe that DCT-based upsampling holds significant potential in deep learning applications. We consider moving Theorem 5.2 to the appendix while emphasizing its potential in the main paper. In summary, we will refine the writing in these two paragraphs by adding proper citations and explicitly highlighting the message we want to deliver in this paper.
>
> **Q3: The metrics and baselines are limited (only FID and UViT), which may not ensure a fair comparison. A broader evaluation with models trained on the same datasets (even if #Parameters, GFLOPs, and training steps differ) would improve the study. Additional metrics, such as Inception Score, could be included.**
>
> In addition to UViT, we have also provided the results on the baseline DiT (please refer to our paper). The reason for choosing these two base modes is that they are well-known models and widely used in the community. Evaluating DCTdiff on more base models is valuable as long as the base model is easy to implement. For example, we have preliminarily implemented DCTdiff using the UNet architecture, the results are promising (Table 11). We believe that DCTdiff is a general method to be applied in different diffusion networks.
>
> Table 11. FID of UNet-based DCTdiff on CIFAR-10 (NFE=100).
>
> | training steps | 200k | 300k | 400k |
> | --- | --- | --- | --- |
> | DCTdiff (UNet) | 5.06 | 4.88 | 4.48 |
>
> Regarding extra metrics, we have added IS, precision, recall, CMMD (as suggested by reviewer VaLT) to evaluate the models. The results are shown in Tables 3, 4, 5.
>
> Table 3. Comparison between UViT and DCTdiff on CIFAR-10 using DDIM sampler (NFE=100)
>
> |  | FID ↓ | CMMD ↓ | IS ↑ | precision ↑ | recall ↑ |
> | --- | --- | --- | --- | --- | --- |
> | UViT | 5.05 | 0.052 | 7.08 | 0.668 | 0.589 |
> | DCTdiff | 4.25 | 0.043 | 7.70 | 0.660 | 0.606 |
>
> Table 4. Comparison between UViT and DCTdiff on FFHQ 128 using DPM-solver (NFE=100)
>
> |  | FID ↓ | CMMD ↓ | IS ↑ | precision ↑ | recall ↑ |
> | --- | --- | --- | --- | --- | --- |
> | UViT | 9.18 | 0.610 | 3.54 | 0.648 | 0.485 |
> | DCTdiff | 6.50 | 0.470 | 3.67 | 0.668 | 0.512 |
>
> Table 5. Comparison between UViT and DCTdiff on AFHQ 512 using DPM-solver (NFE=100)
>
> |  | FID ↓ | CMMD ↓ | IS ↑ | precision ↑ | recall ↑ |
> | --- | --- | --- | --- | --- | --- |
> | UViT (latent) | 10.86 | 0.373 | 11.00 | 0.547 | 0.496 |
> | DCTdiff | 8.76 | 0.335 | 11.00 | 0.632 | 0.496 |
>
> **Q4. Latent diffusion models using VAEs typically produce high-quality images but introduce issues with encoder-decoder invertibility. Hong et al., propose a correction algorithm for this, suggesting that DCTdiff could bypass such issues entirely**
>
> Thank for suggesting the case of encoder-decoder invertibility. We do agree that DCT is a flexible and invertible method of image representation, which differs from the neural network-based image tokenizers. We will include this discussion with paper [1] in the final version of our paper.
>
> [1] Hong, Seongmin, et al. "Gradient-free Decoder Inversion in Latent Diffusion Models." NIPS (2024)

---

> > ### Comment · Reviewer_guDg · 2025-04-04
> >
> > Thanks for the additional experiments and clarifications.
> >
> > For Q1, I think it's a strong point that your method can generate 512×512 samples without using a latent autoencoder. It’s a clear advantage, and I wish the qualitative samples for FFHQ 512 were shared via an anonymous link in the rebuttal.
> >
> > Still, I appreciate the added results. I would like to keep my score as is.

---

### Official Review · Reviewer_5JtJ · 2025-03-11

**Overall Recommendation:** 4

**Summary:**

In this work authors introduce a novel idea to model images in their frequency spaces with diffusion models. Authors show that they can use Diffusion Transformer architectures to model the frequencies of images in a smart way without changing the architecture. There are several new observations on how to achieve this task regarding the need for new scaling etc. With the series of experiments it is shown that the method outperforms pixel-based alternative baseline approach achieving good level of generation quality.

**Claims And Evidence:**

The submission introduces a new method that uses in a brilliant way several observations from compression. Those combinations of known tricks from JPEG with generative modelling are not trivial, and compelling:
- The observation that DCT blocks correspond to VIT patches is clever and insightful
- The zigzag flattening (originally introduce in JPEG) together with reconstruction FID metric that is used for estimating the best trade-off between generation quality and compression rate (speed) introduces great controllability of those two crucial elements in generative modeling with diffusion models.

**Essential References Not Discussed:**

I think the one important reference missing is [1] where authors combine a hierarchical VAE with a diffusion model by using a Discrete Cosine Transform (DCT) to create low-dimensional pseudoinputs, which are then modeled via a diffusion-based prior to efficiently approximate the aggregated posterior and enhance latent space utilization.

[1] Kuzina, Anna, and Jakub M. Tomczak. "Hierarchical VAE with a Diffusion-based VampPrior." Transactions on Machine Learning Research.

**Experimental Designs Or Analyses:**

- The evaluation of upsampling is very limited to comparison with baseline pixel upsampling. This is fine as a proof of concept, but please note that novel diffusion architectures (e.g. DeepFloyd IF) use more advanced approaches also employing several diffusion steps.
- The final evaluation is a comparison to the baseline UViT architecture on a number of different datasets including some high-resolution ones. In those experiments DCTdiff consistently outperforms pixel-space UViT. However, the observed results are far from state-of-the-art (e.g. FID on FFHQ256 2.19 for StyleGAN from 2022 vs 5.08 from this work).

**Methods And Evaluation Criteria:**

The benchmarks used for the evaluation are sufficient in size, authors even include some high-resolution datasets. Nevertheless the method is only compared with the baseline model with the same architecture run in the pixel space. This is fine regarding that the contribution is solely limited to the change of the modeling space, but might be seen as a limitation of the evaluation.

**Other Comments Or Suggestions:**

Small suggestions:
- I don't know how was it defined, but the bold math letters (e.g. x'_y and so on in line 121 right) are hard to read

**Other Strengths And Weaknesses:**

Strengths:
- The proposed method is novel, sound, and makes a lot of sense-
- This work introduces several smart observations from frequency space analysis and compression methods into generative modelling.

Weaknesses:
- The evaluation of the methods is limited to the comparison to the baseline approach in the pixel space

**Questions For Authors:**

- The scaling method introduced in the work is a challenging task. The proposed Entropy-Consistent Scaling is reasonable, but have you considered using different terminal distribution for the diffusion model instead of gaussian? Maybe something like Pareto distribution would be more suitable for the unusual distribution defined by DCT? Such an approach might also help for the SNR Scaling

**Relation To Broader Scientific Literature:**

This submission well discusses the broader scientific literature

**Theoretical Claims:**

There are 3 Theorems with "Sketch of Proofs" in the submission, while there are detailed proofs in the appendix. I did not check the correctness of all proofs, but I am unsure if Theorem 5.1 which discusses connection between pixel-based diffusion and autoregressive modeling is really necessary for this work. I see how it might be a bit relevant to the proposed approach, but it does not provide theoretical grounds for the proposed method.

---

> ### Author Rebuttal · Authors · 2025-04-01
>
> We sincerely appreciate the insightful suggestions and comments provided by the reviewer.
>
> **Q1: The benchmarks used for the evaluation are sufficient in size, authors even include some high-resolution datasets. Nevertheless the method is only compared with the baseline model with the same architecture run in the pixel space**
>
> In addition to the comparison with pixel-based diffusion models, we also compared DCTdiff with the latent diffusion model UViT which utilizes the VAE of StableDiffusion for image compression. Please refer to Table 3 of our paper for the details.
>
> **Q2: Theorem 5.1 might be a bit relevant to the proposed approach, but it does not provide theoretical grounds for the proposed method.**
>
> Theorem 5.1 provides a theoretical foundation for understanding how noise affects different frequency components in the diffusion process. It formalizes the observation that high-frequency signals are more susceptible to noise perturbations due to the energy concentration properties of the DCT. This insight is crucial for justifying why DCTdiff exhibits a faster noise accumulation process than pixel diffusion, ultimately motivating the introduction of the SNR scaling method. We will improve the writing of Section 5.3 to highlight the connection.
>
> **Q3: The evaluation is a comparison to the baseline UViT on a number of different datasets including some high-resolution ones. In those experiments DCTdiff consistently outperforms pixel-space UViT. However, the observed results are far from state-of-the-art (e.g. FID on FFHQ256 2.19 for StyleGAN from 2022 vs 5.08 from this work).**
>
> Thanks for the interesting observation. GANs have been a strong class of models that generate high fidelity images (often measured by FID) (more evaluations can be found in https://paperswithcode.com/sota/image-generation-on-ffhq-256-x-256) yet do not produce diverse images. The main advantages of diffusion models are that they are stable to train and have higher diversity (often measured by recall) than GANs (see papers [1][2]). Our DCTdiff actually achieves better FID than many other diffusion-based models, e.g [3] and [4]).
>
> [1] Dhariwal, Prafulla, and Alexander Nichol. "Diffusion models beat gans on image synthesis." NIPS. 2021.
>
> [2] Boutin, Victor, et al. "Diffusion models as artists: are we closing the gap between humans and machines?." ICML 2023.
>
> [3] Rombach, Robin, et al. "High-resolution image synthesis with latent diffusion models." *CVPR*. 2022.
>
> [4] Kim, Dongjun, et al. "Soft Truncation: A Universal Training Technique of Score-based Diffusion Model for High Precision Score Estimation." ICML. 2022.
>
> **Q4 I think the one important reference missing is [1] where authors combine a hierarchical VAE with a diffusion model by using a Discrete Cosine Transform (DCT) to create low-dimensional pseudoinputs, which are then modeled via a diffusion-based prior to efficiently approximate the aggregated posterior and enhance latent space utilization**
>
> Except for explicit signal compression, it is inspiring to see that DCT can be treated as a low-dimensional latent representation and applied to VampPrior framework to obtain the flexible prior distribution modeled by diffusion. We will add the discussion of [1] to our literature review.
>
> [1] Kuzina, Anna, and Jakub M. Tomczak. "Hierarchical VAE with a Diffusion-based VampPrior." Transactions on Machine Learning Research.
>
> **Q5: Have you considered using different terminal distribution for the diffusion model instead of gaussian? Maybe something like Pareto distribution would be more suitable for the unusual distribution defined by DCT? Such an approach might also help for the SNR Scaling**
>
> We sincerely appreciate the reviewer providing this insightful suggestion. In our DCTdiff work, we follow the Gaussian prior of diffusion due to the fair comparison. But we do think that Pareto distribution which exhibits the power-law (same as image spectral distribution) is very likely to be a great solution for DCT distribution modeling. Moreover, Laplace distribution is also worth exploring for DCT modeling. We will investigate this topic in our next work.

---

> > ### Comment · Reviewer_5JtJ · 2025-04-03
> >
> > Thank you for the rebuttal,
> > Regarding Q3, to clear the misunderstanding, I am familiar with limitations of GANs, and by no means I want to advocate in their favor. What I wanted to highlight is that this work is far from recent state-of-the-art results for image generation. In fact it's even far from such results 3 years ago, as highlighted by FID comparison on FFHQ dataset.
> >
> > I don't have any further questions, and I intend to keep my initial evaluation score

---

### Official Review · Reviewer_vBsQ · 2025-03-13

**Overall Recommendation:** 3

**Summary:**

The paper introduces an end-to-end diffusion modeling framework in the frequency space, instead of in the original pixel space. It shows that the DCT (discrete cosine transform) space could be an effective and near-lossless compression for diffusion modeling, mitigating pixel redundancy and enabling efficient scaling to 512×512 image generation without requiring auxiliary VAEs.

The authors propose a pipeline for token preparation for Diffusion Transformers, accompanied by adjustments to hyperparameters such as noise schedules and loss re-weighting. Experimental results on UViT and DiT architectures show that DCT-based diffusion models outperform pixel-space and latent-space counterparts on FID scores and training efficiency.

**Claims And Evidence:**

- "*suggesting its potential for both discriminative and generative tasks*" (Lines 67-68).
  - The paper only evaluates some generative tasks (unconditional or class-conditional image synthesis), while the capability of DCT space on other generative (e.g. image editing, inpainting and restoration) and discriminative (e.g. image classification and segmentation) tasks is still unknown.
  - Recent studies show that intermediate representations within diffusion networks are effective for discriminative tasks through generative pre-training, observed in both pixel-space (DDAE, arXiv:2303.09769; DDPM-seg, arXiv:2112.03126) and latent-space models (l-DAE, arXiv:2401.14404; REPA, arXiv:2410.06940). Does DCT-based modeling retain similar properties?

- "*outperforms the pixel-based and latent diffusion models regarding generation quality and training speed*" (Lines 62-64).
  - The comparison is restricted to latent diffusion using SD-VAE, an outdated compression model. Many modern image tokenizers (VA-VAE, arXiv:2501.01423; MAETok, arXiv:2502.03444) can also achieve near-lossless reconstruction (rFID < 0.5) and provide compact, diffusion-optimized spaces. Can the DCT space outperform these modern tokenizer specifically designed for "*more diffusible*" latent representations?
  - I understand that complete training on large datasets like ImageNet256x256 is too resource-intensive. However, as mentioned above, recent tokenizer advancements have reduced costs (e.g. 10-20x faster convergence). Therefore, please, as much as you can afford (e.g. even with limited training in a few hundred epochs), provide some results on standard benchmarks like ImageNet-256x256.
  - The baselines also appear under-optimized. For example, I can achieve an FID of 4.5 on unconditional CIFAR-10 with a 100-NFE DDIM sampler, whereas Table 2 reports FIDs of 5.29-6.23. This suggests the pixel- and latent-space models may not be fully converged, undermining the claimed outperformance.

**Essential References Not Discussed:**

The essential related works are mostly discussed.

**Experimental Designs Or Analyses:**

Please refer to the second discussion in "Claims And Evidence" section.

**Methods And Evaluation Criteria:**

The proposed method is technically sound and novel, but its applicability appears limited to ViT-based backbones (e.g. UViT, DiT).
- UNet-based architectures remain competitive, particularly for unconditional and class-conditional image generation on datasets like CIFAR-10, FFHQ, and ImageNet-64 (EDM2, arXiv:2312.02696). Can the proposed DCT-based diffusion be adapted to UNet backbones and also outperform pixel-space counterparts?
- ViT-based architecture, on the other hand, is more preferable when involving text modalities (e.g. text-to-image synthesis) and scaling to larger network capacities. However, the provided evaluations align more closely with UNet strengths rather than ViT advantages.

**Other Comments Or Suggestions:**

Typo: "UViT" in Table 5 should be "DiT".

**Other Strengths And Weaknesses:**

N/A

**Questions For Authors:**

After reading the paper carefully, I acknowledge that the DCT-based space is indeed a more preferable replacement for the raw pixel-space (if the claimed "*potential for both discriminative and generative tasks*" is correct and sound). However, as mentioned above, I doubt whether it is better than modern VAE/AE-based image tokenizers. Those tokenizers compress images with higher compression ratio, offering a more compact and "*diffusible*" latent space for diffusion modeling, and may optionally preserve semantic information for unified generation-and-understanding tasks, particularly for multi-modal models. Btw, I think training an auxiliary tokenizer on the DCT space (instead of pixel-space) may also sound reasonable.


**2025/04/04: [Replying to Reply Rebuttal Comment by Authors]**:

Thank you for the additional discussion.

Regarding the performance and efficiency comparison, I acknowledge the authors' point that evaluating DCT-space designs against latent-space sampling involves complexities influenced by factors such as model size, patch size, and NFEs. The revised Tables 2a & 2b offer improved clarity. For the final version, I suggest incorporating more direct visualizations, similar to the performance trade-off curves (e.g., Fig. 1, 8, 9 presented in EDM2 [arXiv:2312.02696]), to further enhance the presentation.

I also accept the explanation that the implementation is particularly well-suited for Transformer-based models. The current UNet-based comparison, while noted as incomplete, is acceptable and does not critically impact my overall assessment.

Regarding the UViT baselines, I am glad the authors confirm that the reimplementation is strong and valid. The patch_size in UViT on 512x512 is indeed 4 (instead of the common practice ps=2 in modern architectures). I apologize for the false alarm.

Based on the clarifications provided, I am raising my rating to Weak Accept (actually not very curcial since other reviews are all positive). I strongly recommend that the authors dedicate to reorganizing the results presentation and refining the training details in the final version for improved clarity.

**Relation To Broader Scientific Literature:**

N/A

**Theoretical Claims:**

I do not find any obvious errors in the theoretical analysis.

---

> ### Author Rebuttal · Authors · 2025-04-01
>
> Thank you very much for the detailed and helpful reviews.
>
> **Q1: The paper only evaluates some generative tasks, while the capability of DCT space on other generative and discriminative tasks is still unknown**
>
> We will first rephrase this sentence to avoid any misunderstanding. Our work primarily aims to explore the feasibility of performing diffusion in the DCT space for image generation, with a particular focus on generative capabilities. While we acknowledge the potential discriminative properties of DCT-based modeling, we believe these properties are primarily derived from the generative pre-training mechanism rather than the information format itself. This could also explain why both pixel and latent diffusion models exhibit similar characteristics. We leave the investigation of the discriminative, editing, inpainting and restoration ability of DCT-based diffusion models as future work.
>
> **Q2: The comparison is restricted to latent diffusion using SD-VAE, an outdated compression model, Can the DCT space outperform these modern tokenizer?**
>
> First, we will rewrite this claim by clarifying that the latent diffusion is SD-VAE based, making the scope of our claim clearer. Second, Our goal is not to beat these emerging tokenizers with dedicated designs, we explore how far the image diffusion modeling can go without a pretrained tokenizer. Moreover, our work does not conflict with image tokenizers, e.g. we totally agree that exploring DCT tokenizer is promising, and the technique of these new tokenizers can also be used to develop DCT tokenizer.
>
> **Q3: Can you provide some results on ImageNet-256?**
>
> Given the limited GPUs, we have prioritized our experiments on the other tasks: scaling up, extra evaluations, and explorations of UNet-based DCTdiff.
>
> **Q4: The baseline on CIFAR-10 appears under-optimized**
>
> For fair comparison, our implementation of UViT and DCTdiff on CIFAR-10 used patch size (ps=4) (different from ps=2 in original UViT paper). If we apply ps=2 on DCTdiff, block size will be 1, yielding only the DC frequency and preventing frequency reweighting. To address your concern, we have implemented another baseline of UViT using ps=2 (256 tokens). Correspondingly, we increase the model size of DCTdiff to ensure the same computational complexity for a single network forward pass. Results on Tables 9 and 10 show that DCTdiff is significantly better than UViT in terms of FID. We acknowledge that the FID 5.05 on the UViT is still higher than the 4.5 you tested (might be due to the warmup steps, machine and software difference etc.), but our code and all trained models will be released for reproducibility.
>
> Table 9. Extra CIFAR-10 benchmark, FID-50k using DDIM sampler
>
> |  | 100 | 50 | 20 | 10 |
> | --- | --- | --- | --- | --- |
> | UViT (small, 256 tokens) | 5.05 | 6.24 | 17.83 | 73.05 |
> | DCTdiff (mid_deep, 64 tokens) | **4.25** | **4.54** | **5.96** | **11.17** |
>
> Table 10. Extra CIFAR-10 benchmark, FID-50k using DPM-solver
>
> |  | 100 | 50 | 20 | 10 |
> | --- | --- | --- | --- | --- |
> | UViT (small, 256 tokens) | 4.82 | 4.85 | 4.92 | 10.72 |
> | DCTdiff (mid_deep, 64 tokens) | **4.40** | **4.43** | **4.56** | **8.82** |
>
> **Q5: Can the proposed DCT-based diffusion be adapted to UNet backbones and also outperform pixel-space counterparts?**
>
> It is an interesting question. Although we think that Transformer has many advantages than UNet regarding the image DCT implementation: ease of (1) Cb Cr 2x subsampling, (2) elimination of high frequencies, and (3) frequency loss reweighting, we have investigated the possibility of UNet-based DCTdiff using the ADM codebase. Concretely, we remove the Cb Cr 2x subsampling and loss reweighting in the preliminary experiment, and just convert the RGB channel to YCbCr followed by DCT transform, the resulting frequencies yield the input tensor with shape (32, 32, 3) on CIFAR-10. We trained the UNet-based DCTdiff for 400k steps, the FID is shown in Table 11 and the results are promising. We believe further exploration of UNet-based DCTdiff will lead to better generation quality.
>
> Table 11. FID of UNet-based DCTdiff on CIFAR-10 (NFE=100).
>
> | training steps | 200k | 300k | 400k |
> | --- | --- | --- | --- |
> | DCTdiff (UNet) | 5.06 | 4.88 | 4.48 |
>
> **Q6: ViT-based architecture is more preferable when scaling to larger network capacities**
>
> We provide the scaling experiments below.
>
> Table 6. FID on CIFAR-10 using DDIM sampler, ps=4
>
> |  | NFE=100 | NFE=50 | NFE=20 |
> | --- | --- | --- | --- |
> | UViT (small) | 7.25 | 8.45 | 21.18 |
> | DCTdiff (small) | 6.51 | 6.62 | 7.87 |
> |  |  |  |  |
> | UViT (mid) | 6.23 | 7.88 | 20.48 |
> | DCTdiff (mid) | 5.02 | 5.21 | 6.81 |
> |  |  |  |  |
> | UViT (mid, deep) | 6.05 | 7.33 | 20.27 |
> | DCTdiff (mid, deep) | 4.25 | 4.54 | 5.96 |
>
> Table 7. FID on FFHQ 128 using DPM sampler
>
> |  | NFE=100 | NFE=50 | NFE=20 |
> | --- | --- | --- | --- |
> | DCTdiff (small) | 6.50 | 6.55 | 7.72 |
> | DCTdiff (mid) | 5.13 | 5.20 | 6.19 |
> | DCTdiff (mid, deep) | 4.98 | 5.05 | 5.94 |

---

> > ### Comment · Reviewer_vBsQ · 2025-04-02
> >
> > Thank you for providing a detailed rebuttal and addressing many of the points raised in the initial reviews. I appreciate the effort, particularly the additional evaluations and the preliminary investigation into UNet-based DCTdiff.
> >
> > However, several key concerns regarding the experiments persist, which prevent me from fully endorsing the paper at this stage.
> > - Table 2 indicates that under common sampling configurations (e.g., NFE=100-250, as often used for ADM and DiT), DCTdiff does not appear to offer a speed advantage compared to Latent UViT and can, in fact, be considerably slower.
> > - While I am grateful for the UNet-based results in Table 11, the evaluation lacks a comparison against a standard UNet baseline. The reported FID=4.48 (at 400k) seems worse compared to common baselines (e.g., the official DDIM paper reported FID=4.16 with NFE=100, for a 35.7M DDPM Ho's UNet after 400k * 256 / 50000 = 2000 epochs).
> > - Concerns regarding the under-optimization of the baselines used still linger, particularly in Tables 6 and 9. For example, based on common results, a standard UViT (patch_size=2, 44M #params) trained for 1200 epochs on CIFAR-10 should readily achieve an FID_50k around 4.5 using 100 DDIM steps. The reported FIDs appear significantly weaker than this.
> > - Alignment with common practices: There seem to be instances where experimental configurations for baselines might deviate from common practices. For example, the response to Reviewer VaLT (Q2) mentions using 256 tokens for the 512x512 UViT baseline. This implies a patch_size of 4 (512 / 8 downfactor / 4 patchsize = 16, 16x16=256 token), which represents a considerably lower downsampling rate than typically used in latent UViT/DiT (downfactor=8, patch_size=2). Ensuring fairness in comparisons is vital, aligning not only #params but also these choices (like patch sizes, downsampling rates) with standard practices. It is currently challenging for the reader to judge the fairness and significance of the reported gains.
> >
> > In conclusion, while I appreciate the novelty of exploring diffusion models in the DCT space, the highlighted experimental limitations regarding sampling speed, UNet performances, reported baselines, and overall experimental fairness prevent me from raising my score.
> >
> > Therefore, I maintain my score of Weak Reject. However, I acknowledge the paper's interesting direction and the authors' constructive engagement. I would not strongly object to its acceptance.

---

> > > ### Author Response · Authors · 2025-04-03
> > >
> > > We are glad to hear that some of the concerns of the reviewer have been addressed, and we thank you for recognizing the novelty of our work. To address the remaining concerns, we would like to add one more discussion along with a kind reminder of **the factual errors the reviewer had** in the above comments.
> > >
> > > **Q1: The inference time of DCTdiff is slower than latent UViT on 512x512 benchmark in the large NFE condition.**
> > >
> > > To again highlight our fairness, we report the wall-clock time without considering their sampling quality (we surely admit the results). However when considering inference time at comparable generation quality, our DCTdiff demonstrates clear advantages: **latent UViT requires 20 mins to achieve FID 10.89, whereas DCTdiff achieves FID 8.04 in just 9 mins on FFHQ 512.**
> > >
> > > Table 2a. Wall-clock inference time and FID on **AFHQ** 512.
> > >
> > > |  | NFE=100 | NFE=50 | NFE=20 | NFE=10 |
> > > | --- | --- | --- | --- | --- |
> > > | UViT (latent) | **20m 14s (FID 10.86)** | 13m 24s (FID 10.86) | 9m 18s (FID 11.94) | 7m 57s (FID 28.31) |
> > > | DCTdiff | 47m 50s (FID 8.76) | 23m 53s (FID 8.87) | **9m 34s (FID 10.05)** | 4m 47s (FID 21.05) |
> > >
> > > Table 2b. Wall-clock inference time and FID on **FFHQ** 512.
> > >
> > > |  | NFE=100 | NFE=50 | NFE=20 | NFE=10 |
> > > | --- | --- | --- | --- | --- |
> > > | UViT (latent) | **20m 14s (FID 10.89)** | 13m 24s (FID 10.94) | 9m 18s (FID 11.31) | 7m 57s (FID 23.61) |
> > > | DCTdiff | 47m 50s (FID 7.28) | 23m 53s (FID 7.09) | **9m 34s (FID 8.04)** | 4m 47s (FID 19.67) |
> > >
> > > **Q2: The UNet-based DCTdiff (initial trial) underperforms the UNet pixel diffusion**
> > >
> > > As requested by the reviewer, we conducted a preliminary exploration of the UNet-based DCTdiff during the rebuttal period, with the aim of evaluating its feasibility. As mentioned, we did not implement (1) Cb Cr 2x subsampling and (2) frequency loss reweighting due to the inherent constraints posed by the fixed input shape of UNet and regular Conv kernel (see the table below). This also justifies our decision to explore Transformer-based models for DCTdiff, with the approach acknowledged by reviewer 5jtj as clever and insightful. It is important to note that our initial experiment was intended solely to assess the viability of DCTdiff with UNet, rather than to achieve optimal performance. Unlike the reviewer’s immediate assessment, we believe that exploration in this direction remains promising with dedicated designs for the input space and the application of dilated convolution kernels.
> > >
> > > Implementation comparison of DCTdiff on Transformer and UNet.
> > >
> > > |  | Transformer | UNet |
> > > | --- | --- | --- |
> > > | Cb Cr 2x subsampling | easy | difficult |
> > > | frequency loss reweighting | easy | difficult |
> > > | elimination of high frequencies | easy | difficult |
> > >
> > > **Q3 and Q4: The reviewer raises concerns about the optimization of the UViT baseline and questions the fairness of our experiments, noting that our reimplementation yields a slightly worse FID on CIFAR-10 compared to his/her reported 4.5.**
> > >
> > > - To ensure a fair comparison, we evaluated UViT and our DCTdiff using the same parameter settings as recommended in the UViT paper. We found that using batch_sz=256 yields much better FID on CelebA 64 dataset than the original UViT using batch_sz=128 (**our reimplementation received FID 1.57, much better than the 2.87 reported in UViT paper**). **This demonstrates the high quality of our baseline implementation**. Given this finding, we use batch_sz=256 in all benchmarks except for ImageNet and 512x512 datasets (see Table 6 of our paper). We have confirmed that the difference on CIFAR-10 (5.05 vs. 4.5) is caused by the batch size setting (128 vs. 256). But we used the same batch_sz=256 for both UViT and DCTdiff for fair comparison. Once again, we emphasize that our code and ckpts will be fully released for reproducibility. We would appreciate it if the reviewer could revisit all our experimental settings (detailed in Table 6 of our paper) regarding the fair comparison with UViT.
> > > - Notably, on CIFAR-10, our DCTdiff (FID 4.25) still outperforms the baseline (FID 4.5) reported by the reviewer.
> > > - The official UViT implementation for 512x512 datasets is indeed 256 tokens (downfactor=8, patch_size=4), **not the one mentioned by the reviewer** (downfactor=8, patch_size=2). Please check https://github.com/baofff/U-ViT/blob/main/configs/imagenet512_uvit_large.py)

---

### Official Review · Reviewer_VaLT · 2025-03-13

**Overall Recommendation:** 4

**Summary:**

The paper propose DCTDiff that models images in the discrete cosine transform (DCT) space. The paper discusses the design space of DCTdiff and reveals interesting properties of image modeling in the DCT space such as spectral autoregression nature of pixel diffusion models.

**Claims And Evidence:**

The paper claims that "DCT Upsampling Outperforms Pixel Upsampling". However, it is only compared against interpolation methods and not any super resolution methods.

Table 4 shows that the training cost for DCTdiff is lower than UViT in terms of GFLOPs. However, the actual wall clock time will be different due to GPU optimizations. What is the inference time comparison for the two models?

The gap in the performance difference between the two methods diminishes as the NFE increases, especially for class conditional generation (Table 2). So, the advantage of the proposed method is not clear when scaling the test time compute.

**Essential References Not Discussed:**

The results do not compare against the prior work DCTransformer [1] that also uses DCT space for image modeling. How is the proposed method better? A detailed comparison of the similarities and differences with [1] will elucidate the advantages of the proposed DCTdiff.

[1] Charlie Nash, Jacob Menick, Sander Dieleman, and Peter W Battaglia. Generating images with sparse representations. ICML, 2021

**Experimental Designs Or Analyses:**

Missing experiments on the scalability of the proposed method. Does the performance improve with increasing size of the model?

**Methods And Evaluation Criteria:**

The proposed method is quantitatively evaluated only using FID metric. How about Inception Score and Precision/Recall [1] scores?
There have been better metrics proposed in recent years such as CMMD [2]. How about the performance comparisons using CMMD?

[1] Karras, T., Laine, S., and Aila, T. A style-based generator architecture for generative adversarial networks. CVPR, 2019
[2] Sadeep Jayasumana, Srikumar Ramalingam, Andreas Veit, Daniel Glasner, Ayan Chakrabarti, Sanjiv Kumar, Rethinking FID: Towards a Better Evaluation Metric for Image Generation, CVPR 2024

**Other Comments Or Suggestions:**

Typo - Table 5 should be DiT

**Other Strengths And Weaknesses:**

The paper does not discuss the limitations of the proposed method.

**Questions For Authors:**

I would like to see answers specifically to the following questions in the rebuttal.

1. How is the proposed method compared to closely related DCTransformer? Is training a DiT-based model with the dense DCT image from DCTransformer better?

2. How about performance comparisons using Inception Score. Precision/Recall scores and CMMD metric?

3. What is the wall clock inference time comparison?

**Relation To Broader Scientific Literature:**

The key contribution of the method is to use DCT space for image modeling instead of the pixel space. However, prior work has already shown that DCT space is effective for image generation and so the advantage of the proposed method over the prior work is not clear.

**Theoretical Claims:**

Yes, no major concerns

---

> ### Author Rebuttal · Authors · 2025-04-01
>
> We thank the reviewer for the valuable comments which help improve the paper.
>
> **Q1: The paper claims that "DCT Upsampling Outperforms Pixel Upsampling". However, it is only compared against interpolation methods**
>
> The upsampling we mean in the paper is indeed interpolation. We will make this statement clearer.
>
> **Q2: What is the inference time comparison**
>
> Pixel diffusion and DCTdiff share the same GFLOPs and inference time (Table 1)
>
> Table 1. wall-clock time (NFE=100, 10k samples, one A100)
>
> |  | CIFAR-10 | CelebA 64 | FFHQ 128 |
> | --- | --- | --- | --- |
> | UViT | 2m 54s | 5m 06s | 5m 14s |
> | DCTdiff | 2m 58s | 5m 11s | 5m 12s |
>
> Comparing latent diffusion and DCTdiff, the inference time is different because
>
> - GFLOPs of UViT (latent) = 34*NFE (diffusion) + 1240 (Decoder)
> - GFLOPs of DCTdiff = 133*NFE (diffusion)
>
> The decoder of the VAE is expensive, but DCTdiff has a larger complexity in diffusion as DCTdiff has 1024 tokens and UVIT has 256 tokens. Table 2 shows that DCTdiff is faster in low NFE but is slower in the high NFE condition.
>
> Table 2. Wall-clock inference time on AFHQ 512 (10k samples, A100 GPU). GFLOPs is appended in the brackets.
>
> |  | NFE=100 | NFE=50 | NFE=20 | NFE=10 |
> | --- | --- | --- | --- | --- |
> | UViT (latent) | 20m 14s (4640) | 13m 24s (2940) | 9m 18s (1920) | 7m 57s (1580) |
> | DCTdiff | 47m 50s (13300) | 23m 53s (6650) | 9m 34s (2660) | 4m 47s (1330) |
>
> **Q3: The performance gap diminishes as NFE increases**
>
> In most cases, our DCTdiff has significantly lower FID (20%~30%) than the base model. We believe the small improvement on ImageNet 64 is due to the small model capacity since we have limited GPU to scale on ImageNet. However, we do provide the scaling experiments on other datasets (Table 6,7)
>
> **Q4: How about evaluation using IS, Precision/Recall and CMMD**
>
> Table 3. Comparison between UViT and DCTdiff on CIFAR-10 using DDIM sampler (NFE=100)
>
> |  | FID ↓ | CMMD ↓ | IS ↑ | precision ↑ | recall ↑ |
> | --- | --- | --- | --- | --- | --- |
> | UViT | 5.05 | 0.052 | 7.08 | 0.668 | 0.589 |
> | DCTdiff | 4.25 | 0.043 | 7.70 | 0.660 | 0.606 |
>
> Table 4. Comparison between UViT and DCTdiff on FFHQ 128 using DPM-solver (NFE=100)
>
> |  | FID ↓ | CMMD ↓ | IS ↑ | precision ↑ | recall ↑ |
> | --- | --- | --- | --- | --- | --- |
> | UViT | 9.18 | 0.610 | 3.54 | 0.648 | 0.485 |
> | DCTdiff | 6.50 | 0.470 | 3.67 | 0.668 | 0.512 |
>
> Table 5. Comparison between UViT and DCTdiff on AFHQ 512 using DPM-solver (NFE=100)
>
> |  | FID ↓ | CMMD ↓ | IS ↑ | precision ↑ | recall ↑ |
> | --- | --- | --- | --- | --- | --- |
> | UViT (latent) | 10.86 | 0.373 | 11.00 | 0.547 | 0.496 |
> | DCTdiff | 8.76 | 0.335 | 11.00 | 0.632 | 0.496 |
>
> **Q5 Missing experiments of model scalability**
>
> We performed scaling experiments on CIFAR-10 and FFHQ 128 during the short rebuttal period.
>
> Table 6. FID-50k on CIFAR-10 using DDIM sampler, patch_sz=4
>
> |  | NFE=100 | NFE=50 | NFE=20 |
> | --- | --- | --- | --- |
> | UViT (small) | 7.25 | 8.45 | 21.18 |
> | DCTdiff (small) | 6.51 | 6.62 | 7.87 |
> |  |  |  |  |
> | UViT (mid) | 6.23 | 7.88 | 20.48 |
> | DCTdiff (mid) | 5.02 | 5.21 | 6.81 |
> |  |  |  |  |
> | UViT (mid, deep) | 6.05 | 7.33 | 20.27 |
> | DCTdiff (mid, deep) | 4.25 | 4.54 | 5.96 |
>
> Table 7. FID-50k on FFHQ 128 using DPM sampler
>
> |  | NFE=100 | NFE=50 | NFE=20 |
> | --- | --- | --- | --- |
> | DCTdiff (small) | 6.50 | 6.55 | 7.72 |
> | DCTdiff (mid) | 5.13 | 5.20 | 6.19 |
> | DCTdiff (mid, deep) | 4.98 | 5.05 | 5.94 |
>
> **Q6 Compare DCTransformer with DCTdiff**
>
> DCTdiff differs from DCTransfomer in several key aspects: probabilistic modeling, image representation, network and image tokenization. We summarize their differences in Table 8. Overall, DCTdiff exhibits a straightforward way for image frequency generative modeling. Performance-wise, DCTdiff achieves FID 7.28 on FFHQ while DCTransformer has FID 13.06.
>
> The only overlap between DCTransformer and DCTdiff is using the DCT transform and YCbCr color transform. But these well-known techniques of JPEG compression cannot be credited to DCTransformer.
>
> Table 8. Differences between DCTransformer and DCTdiff
>
> |  | DCTransformer | DCTdiff |
> | --- | --- | --- |
> | Probability modeling | Autoregression (each conditional distribution is further factorized into 3 distributions, see their Eq(2)) | Diffusion |
> | Image representation | tuples (channel, position, value) | Y, Cb, Cr |
> | Network | a Transformer with 1 encoder and 3 decoders used for predicting channel, position, value, respectively | a single diffusion Transformer |
> | use quantization | yes (has information loss) | no |
> | DCT block size | fixed (8/16) | flexible for different resolution generation |
>
> **Q7: What is the limitation of this paper?**
>
> The limitation is that we did not explore other generative applications (e.g. image inpainting) and discriminative tasks. Also, frequency-oriented Transformer architecture and super-resolution image generation were not covered in this paper. We think these are promising directions for future work.

---

> > ### Comment · Reviewer_VaLT · 2025-04-03
> >
> > Thanks to the authors for the rebuttal. It addressed some of my concerns.
> >
> > **DCT Upsampling**: How about comparison to super resolution methods in term of quality and complexity? Why was it not compared?
> >
> > **Scaling**: What is the complexity of the small, mid and mid-deep models? It is difficult to compare the results without knowing the complexity of the three models.
> >
> > **Inference time**: Table 2 shows that DCTdiff has slower inference time due to the larger FLOPs resulting from the higher resolution (more tokens). But it has better quality in terms of FID as mentioned in the reply to reviewer vBsQ. However, a more fair comparison is using a larger model for UViT with the same NFE and having similar inference time. Is DCTdiff better than using a larger UViT model?
> >
> > **Comparison with DCTransformer**: Is the quantitative comparison on FFHQ between the two models of same complexity?
> >
> > **Limitations**: The authors mentioned the limitations of the paper in terms of experiments but I was referring to the limitations of the method. Is using DCT transform always better for image generation? Are there any tasks where DCT based diffusion is not better than using pixel based diffusion?

---

> > > ### Author Response · Authors · 2025-04-04
> > >
> > > **Q1: DCT Upsampling: compare it with super-resolution methods**
> > >
> > > Initially, we indeed think that DCT upsampling can be applied in the cascade diffusion for super-resolution generation, For example, ADM generates a 512x512 image by first generating a 128x128 image, then using pixel interpolation to get the coarse 512x512 image, it is finally refined by a super-resolution model. We had tried to replace the pixel interpolation with DCT upsampling in this approach. However, the ckpt loading of the super-resolution model (released by ADM) was problematic, which prevented us from further implementation. We will continue trying to solve this issue and add the super-resolution results in the final version of our paper.
> > >
> > > **Q2: What is the complexity of the small, mid and mid-deep models**
> > >
> > > Due to the 5000-character limitation, we had to leave out these details. Now, they are shown in Table 6a.
> > >
> > > Table 6a. Model parameters and training GFLOPs on CIFAR-10. UViT and DCTdiff share the same settings and GFLOPs.
> > >
> > > |  | hidden_dim | depth | #para | GLOPFs |
> > > | --- | --- | --- | --- | --- |
> > > | UViT (small) | 512 | 12 | 44M | 2.87 |
> > > | UViT (mid) | 768 | 16 | 130M | 8.45 |
> > > | UViT (mid-deep) | 768 | 20 | 161M | 10.44 |
> > >
> > > **Q3: A fairer comparison is using a larger model for UViT with the same NFE and having similar inference time**
> > >
> > > Thanks for the insightful suggestion. From Table 2, we first know that the inference time and GFLOPs are strongly correlated. To answer your question, we can compare the FID of the UViT and DCTdiff under the same GFLOPS and NFE. Although training a larger UViT on 512x512 is time-consuming, we can still do the comparison on CIFAR-10 (Table 6a). The results show that DCTdiff receives better FID than UViT under the same NFE and GFLOPs.
> > >
> > > Table 2. Inference time on AFHQ 512 (10k samples). GFLOPs is in the brackets.
> > >
> > > |  | NFE=100 | NFE=50 | NFE=20 | NFE=10 |
> > > | --- | --- | --- | --- | --- |
> > > | UViT (latent) | 20m 14s (4640) | 13m 24s (2940) | 9m 18s (1920) | 7m 57s (1580) |
> > > | DCTdiff | 47m 50s (13300) | 23m 53s (6650) | 9m 34s (2660) | 4m 47s (1330) |
> > >
> > > Table 6a. FID on CIFAR-10 using DDIM sampler. Inference GFLOPs is in the bracket.
> > >
> > > |  | NFE=100 | NFE=50 | NFE=20 |
> > > | --- | --- | --- | --- |
> > > | UViT (small) | 7.25 (287) | 8.45 (143) |  21.18 (57) |
> > > | DCTdiff (small) | 6.51 (287) | 6.62 (143) | 7.87 (57) |
> > > |  |  |  |  |
> > > | UViT (mid) | 6.23 (845) | 7.88 (422) | 20.48 (169) |
> > > | DCTdiff (mid) | 5.02 (845) | 5.21 (422) | 6.81 (169) |
> > > |  |  |  |  |
> > > | UViT (mid, deep) | 6.05 (1044) | 7.33 (522) | 20.27 (209) |
> > > | DCTdiff (mid, deep) | 4.25 (1044) | 4.54 (522) | 5.96 (209) |
> > >
> > > **Q4: Comparison with DCTransformer**
> > >
> > > From Table 3 of the paper DCTransformer, we know that DCTransformer used a much larger network (473M parameters) than our DCTdiff (130M) on FFHQ benchmark. It is not surprising to us because DCTransformer applies 1 encoder + 3 decoders, while our DCTdiff only uses 1 decoder-only architecture.
> > >
> > > **Q5: Method limitations**
> > >
> > > The only case where DCTdiff did not outperform pixel diffusion (UViT) is CelebA 64, as we have shown and discussed in our paper. Regarding the limitation of our method, our most important finding is that image DCT modeling has the challenge of dealing with the ‘power-law’ (low frequencies have much larger magnitudes than the high frequencies). This power-law property pushes us to propose our entropy-consistency scaling and SNR scaling (deal with the low energy of high frequencies) for later Gaussian perturbation. By contrast, pixel diffusion does not have these extra two operations. However, as mentioned by reviewer 5JtJ, replacing the Gaussian prior with a Pareto prior might be a better choice for DCT diffusion modeling because image frequency and Pareto share a similar ‘power-law’ distribution. Applying Perato potentially enables us to remove the operation of entropy-consistency scaling and SNR scaling, and likely yields a better generation quality.
> > >
> > > As highlighted in the title of our paper, we aim to deliver the message to the researchers that image modeling in the frequency domain is truly promising. While some recent works [1] [2] have noticed the potential of spectrum, spectral image modeling still lacks sufficient attention compared to traditional pixel modeling. We plan to continue investigating the image DCT modeling and hopefully bring more insights to the community.
> > >
> > > [1] Frequency Autoregressive Image Generation with Continuous Tokens." arXiv:2503.05305.
> > >
> > > [2] NFIG: Autoregressive Image Generation with Next-Frequency Prediction." arXiv:2503.07076.
> > >
> > > **Q6: Results on ImageNet 64 by scaling from small to mid**
> > >
> > > In our initial rebuttal, we said that ‘We believe the small improvement on ImageNet 64 is due to the small model capacity. We have now verified our hypothesis by scaling the model from small to mid. The resulting FID comparison is 4.69 (DCTdiff, mid) vs. 5.85 (UViT, mid). Full results will be added to our paper.
> > >
> > > We hope our responses have addressed your concerns and doubts

---

### Decision · Program_Chairs · 2025-05-01

**Decision:**

Accept (poster)

**Comment:**

This paper proposes a DCT space based diffusion model design, and show that it yields strong results without having to rely on pretrained latent codes. It also provides theoretical analysis, linking it to the comparison between diffusion and autoregressive modeling in frequency space. After extensive discussions, reviewers unanimously agree that this is a strong submission, which the AC agrees with.